# THE ENTITY-DEDUCTION ARENA 🏛: A PLAYGROUND FOR PROBING THE CONVERSATIONAL REASONING AND PLANNING CAPABILITIES OF LLMS

## ABSTRACT

Large language models (LLMs) are currently effective at answering questions that are clearly asked. However, when faced with ambiguous queries they can act unpredictably and produce incorrect outputs. This underscores the need for the development of intelligent agents capable of asking clarification questions to resolve ambiguities effectively. This capability requires complex understanding, state tracking, reasoning and planning over multiple conversational turns. However, directly measuring this can be challenging. In this paper, we offer a surrogate problem which assesses an LLMs's capability to deduce an entity unknown to itself, but revealed to a judge, by asking the judge a series of queries. This *entity-deducing game* can serve as an evaluation framework to probe the conversational reasoning and planning capabilities of language models. We systematically evaluate various LLMs and discover significant differences in their performance on this task. We find that strong LLMs like GPT-4 outperform human players by a large margin. We further employ Behavior Cloning (BC) to examine whether a weaker model is capable of imitating a stronger model and generalizing to data or domains, using only the demonstrations from a stronger model. We finally propose to use Reinforcement Learning to enhance reasoning and planning capacity of Vicuna models through episodes of game playing, which lead to significant performance improvement. We hope that this problem offers insights into how autonomous agents could be trained to behave more intelligently in ambiguous circumstances.

## 1 INTRODUCTION

In uncertain circumstances, intelligent conversational agents may need to reduce their uncertainty by asking good questions proactively, thereby solving problems more effectively. This requires intricate, interactive, strategic decision-making and reasoning about the agent's next move in a multi-turn conversation. This capability is crucial in various applications, such as multistep task completion, task-oriented chatbots, recommendations, and conversational search.

Traditionally, intelligent behavior has been achieved by modularizing various aspects of such tasks into sub-tasks such as *natural language understanding*, *state tracking*, *planning (policy learning)*, and *response generation*. However, recent advances in LLM-powered systems have made it possible to create an end-to-end pipeline, opening up new possibilities for developing autonomous agents that can complete complex tasks using enhanced planning and memory capabilities. Promising works, such as ReAct (Yao et al., 2022), HuggingGPT (Shen et al., 2023), AutoGPT (Significant Gravitas, 2023), LangChain (Langchain-AI, 2023), GPT-Engineer (Anton Osika, 2023) and BabyAGI (Nakajima, 2023), have demonstrated significant potential in this field. These agents require the underlying LLM to retain and recall important information from previous dialogues, resembling the *understanding* and *state tracking* stage. They also rely on the LLM to decompose larger tasks into more manageable components, which is analogous to the *planning* stage. Among them, some approaches (*e.g.*, HuggingGPT) use a *static* planning strategy by first generating the complete plan using the LLM and subsequently tackling each subtask using either the same LLM or alternative models. Other approaches (*e.g.*, AutoGPT) adopt a *dynamic* and *interactive* planning strategy, where the generation of each action is conditioned on the outcome of the previous planning steps.

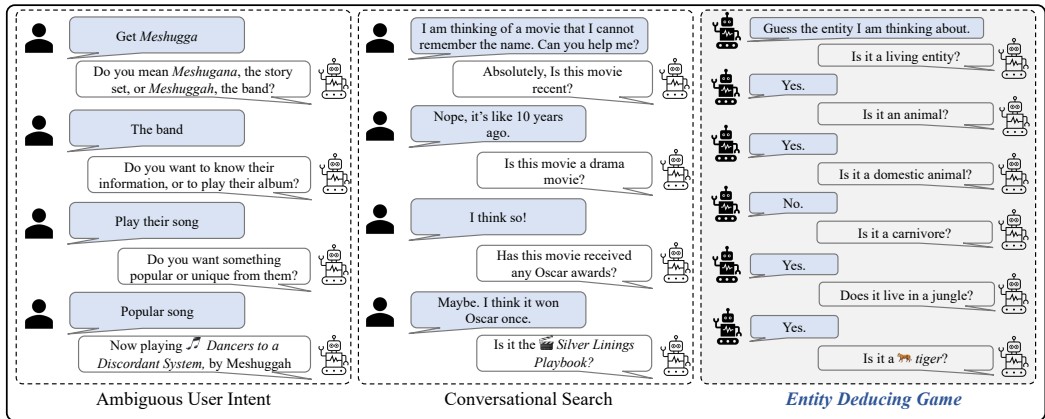

Figure 1: The entity deducing game resembles real scenarios where the agent may need to make strategic decisions regarding the clarification question to be asked based on the current conversation to elicit the actual user intent in as few turns as possible.

While LLM-powered autonomous agents can solve a wide variety of tasks, they can perform poorly when a user's instructions are unclear. This poses a critical challenge – how to accurately elicit and capture the user's intents, which are often nuanced, unpredictable, and noisy, to enable dynamic rather than static human-computer interactions. For example, in Figure 1, the agent needs to accurately assess the current state, and ask strategic questions to steer the conversation to reveal the user's intent. Here, the agent has two objectves: 1) eliminate ambiguity and satisfy the request by gathering the necessary information; 2) ask as few questions as possible.

Progress in this direction is difficult because directly measuring complex understanding, reasoning and planning capabilities is challenging. In this study, we investigate this somewhat overlooked research problem – *how good the LLMs are at asking questions and deducing intent.* We propose to use entity-deducing games, specifically the 20 questions game (Q20) (Akinator, 2007), to assess the complex reasoning and strategic planning capability of LLMs in formulating precise questions/guesses over long conversations (Figure 1). This game requires a model to infer an unidentified entity through a sequence of questions that elicit simple responses of "Yes", "No" or "Maybe" with as few queries as possible. To achieve this, the model must be able to track the dialogue state over turns, and use its reasoning and planning skills to effectively partition and narrow down the search scope.

We systematically evaluated several LLMs for their performance on this task, and found significant differences in their conversational reasoning and planning capabilities. We then investigated a set of research questions to enhance open-source models using demonstrations from high-performing closed-source models. Finally, drawing inspiration from Reinforcement Learning from Human Feedback (RLHF) (OpenAI, 2023), we show how PPO (Schulman et al., 2017) can be used to improve vanilla Vicuna (Chiang et al., 2023) models using game playing directly. Our findings offer insights into potential future advancements in complex reasoning and planning of autonomous agent.

## 2 RELATED WORK

**Evaluation of Complex Reasoning Capabilities** There has been extensive work on evaluating the complex reasoning abilities of LLMs(Huang & Chang, 2022). Prior work has created diverse benchmarks, like HELM (Liang et al., 2022) BIG-bench (Srivastava et al., 2022), SuperGLUE (Sarlin et al., 2020), LAMA (Petroni et al., 2019), and CoT-Hub (Fu et al., 2023), which have enabled researchers to assess LLMs across a spectrum of tasks involving knowledge alignment, commonsense reasoning, long-range coherence and logical deduction, based on various techniques including prompt engineering (Liu et al., 2023) and Chain-of-Thought (CoT) (Wei et al., 2022). As LLMs continue to advance in sophistication, more specialized complex reasoning tasks including arithmetic and math (GSM8K) (Cobbe et al., 2021), commonsense (StrategyQA, ARC) (Geva et al., 2021; Clark et al., 2018), symbolic (BBH) (Suzgun et al., 2022), knowledge (MMLU) (Hendrycks et al., 2020), coding (HumanEval) (Chen et al., 2021), and factual (SummEdits) (Laban et al., 2023) have been developed to gauge their real-world abilities.

**Evaluation of Planning** Evaluation of the planning abilities of LLMs is relatively rare. Valmeekam et al. (2022) proposed an assessment framework to gauge the planning capabilities of LLMs to generate valid actions to achieve a special goal and provide a rationale for the plan. Their evaluation on 8 planning tasks reveals LLMs, like GPT-3 (Brown et al., 2020), seem to display a dismal performance. Valmeekam et al. (2023) further evaluates on GPT-4 and suggests the autonomous learning capacity of LLMs to formulate plans is limited and dependent on properly designed heuristics. Xie et al. (2023) similarly indicate that LLMs may encounter difficulties in generating planning goals related to numerical or spatial reasoning, while being responsive to the specific prompts used. Unlike these studies, our task solely relies on textual representations of goals, and focus on evaluating LLMs under multiturn conversational scenarios.

**Multiturn benchmarks** MT-Bench (Zheng et al., 2023) assesses the multi-turn conversation and instruction-following ability of LLMs by annotating their responses to over 80 challenging questions involving writing, role-play, extraction, reasoning, math, coding and knowledge. Bang et al. (2023) evaluates LLMs on 23 different datasets encompassing tasks related to logical reasoning, non-textual reasoning, and commonsense reasoning. The study reveals that incorporating interactive features in LLMs can enhance their performance by employing multi-turn prompt engineering.

**Entity-deduction game** Testing the model's ability to deduct an ambiguous entity or asking clarification questions (Aliannejadi et al., 2019; Cho et al., 2019) has been utilized as a testbed on dialogue systems and visual reasoning tasks. InfoBot (Dhingra et al., 2016) uses reinforcement learning to learn an optimal dialog policy for identifying movie entries from a movie database. ESP (Von Ahn & Dabbish, 2004) and Peekaboom (Von Ahn et al., 2006) demonstrated that deduction games can effectively gather labeled data. GuessWhat?! (De Vries et al., 2017) and ReferIt (Kazemzadeh et al., 2014), assess the visual reasoning capabilities of tested models. These benchmarks involve a model seeing an entity-deduction conversation based on an image to guess the referred object in the image. Our work instead aims to gauge on the model's ability on generating the conversation.

## 3 ENTITY-DEDUCTION ARENA (EDA)

During an entity-deducing game session, two players engage in a collaborative game regarding a specific entity. One player, the "*judge*" (J), is provided with the entity and is expected to respond to queries or guesses from the guesser using only the responses "Yes," "No," or "Maybe,". The other player, referred to as the "*guesser*" (G) is agnostic about the entity, and is supposed to pose a series of questions to deduce the entity using as few queries as possible.

The judge does not require access to the dialogue history and only needs the entity and current question to provide a response [1]. This task is akin to closed-book QA (Roberts et al., 2020), which state-of-the-art LLMs can reasonably handle. On the other hand, playing the guesser is more demanding. A proficient G necessitates several multi-turn dialogue capabilities working in synergy:

- **State tracking and understanding**: G must possess a comprehensive understanding of the ongoing dialogue to determine which questions have been asked and which ones have not. They also need to discern their current position in the game and display proficiency in comprehending multi-turn context and resolving coreference.
- **Strategic planning:** Given the current state, G needs to devise a policy for strategically asking questions that can most likely progress towards a better state. They must ensure that questions are asked sparingly to avoid redundant queries that yield minimal information gain. Additionally, the questions should be consistent with prior acquired knowledge.
- **Inductive reasoning:** Finally, G needs to employ their entire conversation comprehension to generate a conjecture based on their own knowledge that satisfies all the conditions acquired during the game. This task is more challenging than simply verifying if a given entity fulfills all conditions, as the enumeration of the entire entity space is typically intractable. Typically, these tasks involve the development of a *hierarchical representation* or a balanced index tree to facilitate retrieval. LLMs must inherently establish such a *taxonomy representation* to efficiently and accurately identify the correct entity.

---

[1]Our experiment indicate that incorporating entire dialogue history negatively impacts accuracy. The additional information tends to confuse the judge rather than improve understanding.

Our assessment, referred to as the *Entity-Deduction Arena* (EDA), focuses on evaluating the capability of various LLMs as the guesser, as a proxy to probe their overall capabilities in handling complex multi-turn conversational tasks involving proactively asking clarification questions.

## 3.1 EXPERIMENTAL SETTINGS

**Datasets** We conducted the evaluation on two proposed datasets: *Things* and *Celebrities*. The *Things* dataset comprises a total of 980 entities that have been manually curated. The entities are classified into categories such as objects, animals, and foods. For the purpose of evaluation, we have set aside a random selection of 30 items, while the remaining 950 entities are used for training. As for the *Celebrities* dataset, it consists of 98 celebrity names from different nationality, eras of life and various occupations including sports, entertainment, politics. The composition of each dataset is provided in Appendix A. We use 30 examples for evaluation and 68 examples for training.

**Judge (J)** We employ GPT-3.5-turbo as the judge. The judge takes the entity, questions from the guesser and the following prompt to generate a response of "Yes," "No," or "Maybe" for Things dataset. In guessing the celebrity name, the choices are "Yes," "No," or "Dunno". Consequently, the resulting prompt is slightly different. (Appendix B)

> *Based on your knowledge about the entity: {entity}, respond to the following question or guess. Limit your respond to only "Yes.", "No." or "Maybe.", with no explanation or other words. Never say the answer in your response. If the question is to solicit the answer, respond "No.". \n Question/Guess: {question} (Yes/No/Maybe)*

Whenever the correct answer is contained in the generation from G as an exact substring match, we manually set the output of J to be *"Bingo!"* and G wins this game. At the penultimate step of J, an extra prompt *"You must guess now, what's it?"* will be appended to J's response to guide G in making the ultimate guess. To emulate more deterministic responses from J, we use a temperature of 0.2 for the generation. Admittedly, we observe that the judge model exhibits occasional inaccuracies in its responses, resulting in a noisy environment. It is possible that modifying the prompts could enhance performance. We have not extensively examined this possibility. Nevertheless, this might be a desirable aspect since user responses in real-world scenarios can also be noisy on occasion. This noisy environment of J is consistent with all models and human testers.

**Guesser (G)** The guesser model is agnostic to the entity. It receives the current dialogue history as input and generates the next question or final guess, guided by the instructions provided in the following (the prompt for Celebrities is provide in Appendix B):

> *Your task is to ask a series of questions to deduce the entity that I'm thinking of with as few queries as possible. Only ask questions that can be answered by "Yes," "No," or "Maybe". Do not ask for hint. Make your question brief with no linebreaker. Now start asking a question. \n {dialog history}*

We employed a consistent sampling approach using a temperature of 0.8 for all experiments. The conversation format for each of the assessed LLMs is based on their official guidelines.

**Evaluation metrics** We assess the model's performance by evaluating its final prediction using the Exact Match (EM) criteria [2]. This evaluation considers four key metrics: 1) **#Turns**, which represents the average number of turns taken to complete the game. Games terminate at 20 turns if failed. 2) **Success** rate, which indicates the percentage of games won by G. 3) **#Yes**, which represents the average number of "yes" responses received from the J. 4) **Score**, which is calculated based on a combined game score of **#Turns** and success rate, defined in Eq. (1).

$$S = \begin{cases} 1 - 0.02 \cdot \max(\textbf{\#Turns} - 5, 0) & \text{if G wins,} \\ 0 & \text{if G loses.} \end{cases} \tag{1}$$

---

[2] Alternatively, a more lenient evaluation metric could be employed, *e.g.* embedding similarities or LLM judges. Nevertheless, we discovered that these metrics rely on the specific embedding model or LLM judge employed, potentially resulting in less consistent and reliable evaluations than the EM criteria.

Lower values for **#Turns** and higher values for **Success** and **Score** indicate better performance. The **#Yes** is more of a statistic than a evaluation metric, but we have observed some correlation between this metric and the final performance. Intuitively, a losing game is often characterized by a high frequency of unproductive guesses (with "No" or "Maybe" response from $\mathtt{J}$).

**Human baseline** Collecting static human annotation for this study is a challenging task due to the interactive nature of this research. In order to establish a baseline of human performance, we conducted a large-scale human-in-the-loop study. We set up a game server and recruited 108 human volunteers to interact with the $\mathtt{J}$, and collected a total of 140 and 68 human game play sessions for *Things* and *Celebrities*, respectively. Human guessers were given the same instructions as the LLM guessers and were provided with a tutorial and optional GPT-3.5 retrospection generation for training purposes. Statistics, experimental details and UI are provided in Appendix D.

## 4 BENCHMARKING LLMS ON EDA

| | Things | | | | Celebrities | | | |
|---|---|---|---|---|---|---|---|---|
| | #Turns (↓) | Success (↑) | #Yes | Score (↑) | #Turns (↓) | Success (↑) | #Yes | Score (↑) |
| GPT-4 | **16.9±0.2** | **0.49±0.06** | 6.0±0.2 | **0.40±0.05** | 16.5±0.5 | **0.59±0.04** | 7.3±0.1 | **0.48±0.03** |
| GPT-3.5 | 18.4±0.3 | 0.25±0.04 | 7.1±0.4 | 0.21±0.04 | 17.9±0.3 | 0.41±0.05 | 7.6±0.3 | 0.33±0.04 |
| Claude-2 | 17.6±0.3 | 0.29±0.05 | 4.5±0.3 | 0.25±0.04 | **15.9±0.4** | 0.45±0.06 | 5.3±0.1 | 0.40±0.05 |
| Claude-1 | 18.7±0.1 | 0.15±0.02 | 4.3±0.2 | 0.13±0.02 | 16.7±0.4 | 0.41±0.05 | 4.6±0.2 | 0.35±0.04 |
| Vicuna 13B | 18.7±0.2 | 0.20±0.03 | 5.2±0.3 | 0.17±0.02 | 17.7±0.4 | 0.36±0.08 | 6.8±0.3 | 0.27±0.06 |
| Vicuna 7B | 19.1±0.4 | 0.11±0.06 | 5.7±0.6 | 0.10±0.05 | 19.7±0.3 | 0.05±0.04 | 6.2±0.7 | 0.04±0.03 |
| In-house 65B | 18.1±0.4 | 0.23±0.05 | 5.3±0.2 | 0.20±0.04 | 18.0±0.3 | 0.21±0.03 | 5.4±0.3 | 0.19±0.02 |
| | | | | | | | | |
| V-FT 7B (All) | 18.4±0.2 | 0.20±0.02 | 6.8±0.2 | 0.17±0.02 | 19.0±0.2 | 0.21±0.04 | 9.1±0.3 | 0.16±0.03 |
| V-FT 7B (Things) | 18.5±0.4 | 0.22±0.06 | 6.6±0.2 | 0.18±0.05 | 19.1±1.5 | 0.19±0.20 | 10.3±3.6 | 0.15±0.17 |
| V-FT 7B (Celebs) | 19.7±0.3 | 0.03±0.02 | 1.6±0.1 | 0.03±0.02 | 19.1±0.2 | 0.20 ±0.07 | 7.5±0.6 | 0.16±0.05 |
| V-FT 7B (Suc.) | 18.5±0.5 | 0.28±0.10 | 6.8±0.5 | 0.23±0.08 | 18.6±0.5 | 0.21±0.06 | 7.4±1.4 | 0.17±0.04 |
| V-FT 13B (Suc.) | 18.0±0.5 | 0.29±0.08 | 6.9±0.2 | 0.24±0.07 | 18.6±0.6 | 0.22±0.09 | 7.8±0.5 | 0.18±0.07 |
| | | | | | | | | |
| V-RLGP 7B | 19.3±0.2 | 0.15±0.03 | 3.6±0.1 | 0.12±0.02 | 19.5±0.3 | 0.09±0.05 | 5.8±1.1 | 0.07±0.04 |
| V-RLGP 13B | 17.8±0.2 | 0.31±0.03 | 4.0±0.2 | 0.26±0.02 | 17.5±0.5 | 0.35±0.04 | 6.8±0.2 | 0.29±0.04 |
| | | | | | | | | |
| Human | 18.5±0.5 | 0.24±0.04 | 5.2±0.2 | 0.20±0.04 | 18.1±0.2 | 0.31±0.03 | 7.0±0.3 | 0.25±0.03 |

Table 1: Benchmark of LLMs on the EDA datasets (*Things*, *Celebrities*). **#Turns** denotes the average number of turns taken to complete the game. **Success** denotes the percentage of the games that the guesser model wins. **#Yes** denotes the average number of "yes" response received from the judge. **Score** indicates the reward score defined in Eq. (1). Darker color indicates stronger performance.

We assess several widely-used LLMs, such as GPT-4, GPT-3.5, Claude-1/2, Vicuna (7B-v1.3, 13B-v1.3) (Chiang et al., 2023) using our EDA benchmarks. Unfortunately, we are unable to evaluate Llama-2 or its derivatives due to license restrictions. Llama-1 (Touvron et al., 2023) is also excluded from the comparison as it lacks a conversation finetuned model, rendering its generation not adhering to the instruction. Additionally, we incorporate our in-house LLM (65B) in the comparative analysis. The corresponding results are presented in Table 1. The means and standard deviations for each dataset and LLM were calculated using 30 items from 5 separate runs.

In general, GPT-4 attained the best performance among the models evaluated. Around half of the entities were accurately predicted. The second best model, Claude-2 achieved a significantly lower performance. Interestingly, despite being open-source, the Vicuna 13B model performed comparably well, not trailing too far behind the closed-source models. In particular, Vicuna 13B outperformed Claude-1 when it comes to *Things*. This suggests that open-sourced models have significant potential. Naturally, models with a higher success rate usually require fewer turns.

It is plausible that a small portion of entity deduction game sessions may exist in the webtext and have been included in the pretraining data. However, these sessions likely contribute only a minuscule fraction. We tend to attribute the GPT-4's proficient multi-turn planning and reasoning, particularly during the later stages of the game, to the impressive zero-shot generalization ability of the LLMs.

We present several real game runs in Table 2 on *Things*. Examples on *Celebrities* are in Appendix F. We observe that an effective game strategy typically narrows down the scope through initial questions that divide the remaining search space into two halves of similar capacity, similar to a *binary*

| Successful game | Planning issue (Early Enumeration) |
|---|---|
| G: Is it a living entity? *J: No.* 
 G: Is it man-made? *J: Yes.* 
 G: Can it be held in a single hand? *J: Yes.* 
 G: Is it electronic? *J: Yes.* 
 G: Is it used for communication? *J: Maybe.* 
 G: Can it store information? *J: Yes.* 
 G: Is it a portable device? *J: Maybe.* 
 G: Is it commonly found in homes? *J: Yes.* 
 G: Is it related to entertainment? *J: No.* 
 G: Is it used for work or productivity? *J: Yes.* 
 G: Is it a type of computer or computing device? *J: Maybe.* 
 G: Is it used for creating or editing documents? *J: Yes.* 
 G: Is it a keyboard? *J: No.* 
 G: Is it a mouse? *J: No.* 
 G: Is it a printer? *J: Bingo!* | G: Is the entity a living thing? *J: No.* 
 G: Is the entity an object? *J: Yes.* 
 G: Is the object a tool or a device? *J: Yes.* 
 G: Is the tool or device used for **communication**? *J: No.* 
 G: Is the tool or device used for **transportation**? *J: No.* 
 G: Is the tool or device used for **construction**? *J: No.* 
 G: Is the tool or device used for **entertainment**? *J: No.* 
 G: Is the tool or device used for **cooking**? *J: No.* 
 G: Is the tool or device used for **cleaning**? *J: No.* 
 G: Is the tool or device used for **protection**? *J: No.* 
 G: Is the tool or device used for **sports or athletics**? *J: No.* 
 ... |
| **Planning issue (Redundancy)** | **Reasoning issue (Inconsistency)** |
| G: Is the entity a person? *J: No.* 
 G: Is the entity a living thing? *J: No.* 
 G: Is the entity a thing? *J: Yes.* 
 G: Is the entity an object? *J: Yes.* 
 G: Is the entity a material object? *J: Yes.* 
 G: Is the entity a solid object? *J: Yes.* 
 G: Is the entity a natural object? *J: No.* 
 G: Is the entity an **artificial object**? *J: Yes.* 
 G: Is the entity a **man-made object**? *J: Yes.* 
 G: Is the entity an **inanimate object**? *J: Yes.* 
 G: Is the entity a machine? *J: Yes.* 
 ... | G: Is the entity a living being? *J: No.* 
 G: Is the entity an object or a thing? *J: Yes.* 
 G: Is the object typically found indoors? *J: Yes.* 
 G: Is the object used primarily for a specific purpose? *J: Yes.* 
 G: Is the object small and portable? *J: Maybe.* 
 G: Is the object electronic? *J: Yes.* 
 G: Is the object a device used for communication *J: No.* 
 G: Is the object a device used for **entertainment** or media *J: No.* 
 G: Is the object a device used for productivity or work *J: Yes.* 
 G: Is the object a laptop? *J: No.* 
 G: Is the object a **game console**? *J: No.* 
 ... |

Table 2: The gameplay with the hidden entity "printer". The successful and problematic gameplay scenarios are generated using GPT-4 and vicuna-7B model as G, respectively.

*search* process. During the late game when it becomes challenging to find good binary-partitioning questions, the model will enumerate over a small number of possibilities. Interestingly, the more advanced models like GPT-4 seem to possess this capability to a decent extent, while weaker models exhibit different failure modes in planning and reasoning abilities (Table 2): 1) (Early Enumeration) initiating enumeration too early without first posing high-level questions to narrow down the scope; 2) (Redundancy) asking questions similar to previous ones; 3) (Inconsistency) asking questions or making final guesses that are inconsistent to the answers received in previous turns.

We also compare several games between GPT-4 and human players in Appendix E, highlighting that strong LLMs share similarities with human players in terms of their ability to *backtrack* when they realize they are on the wrong path. LLM may have a superior level of knowledge compared to humans (Appendix E Table 7), empowering them to ask questions using *tacit knowledge* among LLMs. One weakness of the LLMs comparing to humans is that LLMs tend to enumerate or repeat to follow the same pattern of previous turns, demonstrating *self-reinforced* undesirable behaviors. Stronger models occasionally display the ability to escape from this absorbing pattern and rectify early errors, which presumably be attributed to their stronger adherence to the high-level plan. However, the weaker model tends to be attracted to the repetitive patterns or generation artifacts and accumulate errors until they ultimately fail (Appendix F). We then address the following Research Questions (RQs):

**RQ1: How does the LLM make strategies?**    We conducted an analysis to probe the internal dialog states underneath the LLMs, to understand how these states evolve throughout the game, and how do they affect the strategy of the model. To probe the internal state, at each turn before the model asks a question, we prompt the LLM to predict top 5 candidates given the current dialogue history. This probing process is separate from (thus will not affect) the gameplay. The probing prompt can be found in Appendix C. A GPT-4 game session of guessing "guitar" with dialog state is shown in Table 3. A session of guessing "LeBron James" is shown in Table 11 in Appendix C.

From Table 3, it is evident that the model seeks a strategy to ask questions to effectively partition the space among the most promising candidates in the initial 10 turns. After the question is answered, the ranked list of top guesses is adjusted, with certain candidates being eliminated and new candidates being added. For instance, during the fourth turn, the model asks if the item is electronic, which

| Current top 5 guesses before generating question (Ranked) | | | | | Generated question |
|---|---|---|---|---|---|
| computer | cellphone | book | car | house | G: Is it a living entity? *J: No.* |
| book | computer | car | building | smartphone | G: Is it man-made? *J: Yes.* |
| ~~computer~~ | ~~car~~ | ~~building~~ | book | smartphone | G: Can it be held in one hand? *J: Yes.* |
| ~~smartphone~~ | book | pen | key | mug | G: Is it electronic? *J: No.* |
| ~~book~~ | ~~pen~~ | key | mug | deck of cards | G: Is it used for writing or drawing? *J: No.* |
| ~~key~~ | deck of cards | ~~coin~~ | paper clip | dice | G: Is it made of metal? *J: No.* |
| paper book | ~~wooden spoon~~ | plastic toy | glass cup | ~~porcelain plate~~ | G: Is it used in cooking or eating? *J: No.* |
| book | playing card | ~~toothbrush~~ | ~~comb~~ | ~~paper clip~~ | G: Is it used for entertainment or leisure? *J: Yes.* |
| deck of cards | book | puzzle | board game | Rubik's cube | G: Is it associated with a specific sport? *J: No.* |
| book | deck of cards | dice | puzzle | ~~board game~~ | G: Is it used in board games? *J: No.* |
| ~~deck of cards~~ | ~~book~~ | ~~paintbrush~~ | ~~puzzle~~ | ~~dice~~ | G: **Is it associated with music?** *J: Yes.* |
| Harmonica | ~~Guitar pick~~ | Ocarina | Castanets | Recorder | G: Is it a musical instrument? *J: Yes.* |
| ~~Harmonica~~ | ~~Recorder~~ | ~~Kazoo~~ | ~~Maracas~~ | ~~Tambourine~~ | G: **Is it a stringed instrument?** *J: Yes.* |
| Ukulele | ~~Violin~~ | Guitar | Mandolin | ~~Harp~~ | G: Is it commonly used in a symphony orchestra? *J: No.* |
| | | | | | |
| Guitar | ~~Ukulele~~ | ~~Banjo~~ | ~~Mandolin~~ | ~~Lute~~ | G: Is it a guitar? *J: Bingo!* |

Table 3: Probing the top 5 guesses made by GPT-4 during a game of guessing "Guitar". Some guesses were truncated for readability. ~~Striked guesses~~ are eliminated after the question is answered in a retrospective perspective. **Highlighted turns** are the key turns that eliminates all top 5 guesses.

ultimately eliminates the top-ranked guess, "smartphone", after receiving a negative response. The remaining items are shifted one position forward to include a new item "deck of cards" to the top-5.

Interestingly, it is not necessary for the model to always ask questions in order to optimally partition the top-5 candidates. We observed that the strategy also depends on factors including the ease with which a question can be asked to partition the space, and the level of uncertainty the model has about the current top predictions. In situations where the model is uncertain, it may occasionally backtrack and reexamine categories that were previously missed. For instance, during the 11th turn, question was asked that ruled out all of the top candidates. This could be due to the fact that the top items are similar in nature, and the model realize there is a significant proportion of other classes that have been overlooked. The successful questioning in these cases led to the recovery of these overlooked classes.

Additionally, we noticed a high level of consistency in GPT-4's gameplay strategy across multiple repetitions (see example in Appendix K), despite some fluctuation in the order of the questions asked. This suggests that GPT-4 may rely on the dialogue and its own implicit *taxonomy representation* of entities to make decisions, which remains consistent throughout.

**RQ2: Which one is more important in this task, planning or reasoning?** Planning and reasoning abilities affects different stages of game play. Early questions require careful planning to efficiently partition the space, while late game requires more deductive reasoning skills to make an educated guess. We consequently assume that the last turn would only require reasoning ability as no strategic move is needed. With this assumption, we designed the following experiment to investigate the model's planning and reasoning ability in a finer granularity. Given a stronger model GPT-4 and a weaker model Vicuna 7B, and their respective game play trajectories, we only replay the last turn in each trajectory by swapping the guesser model.

|  | **Things (↑)** | **Celebs (↑)** |
|---|---|---|
| GPT-4 | **0.40±0.05** | **0.48±0.03** |
| GPT-4 → Vicuna 7B | 0.12±0.03 | 0.19±0.02 |
| Vicuna 7B | 0.10±0.04 | 0.04±0.03 |
| Vicuna 7B → GPT-4 | 0.11±0.03 | 0.06±0.03 |

Table 4: Ablation on planning and reasoning ability. Numbers are the game scores with 5 repetitions. GPT-4 → Vicuna 7B uses GPT-4 to play all except the last turn, swapping in Vicuna 7B in the last turn. Vicuna 7B → GPT-4 does the opposite.

The results are presented in Table 4 (full table is provided in the Appendix J). Comparing the GPT-4 → Vicuna 7B with Vicuna 7B, we observe that stronger planning ability from GPT-4 results in significant improvement. On the other hand, regarding different reasoners, Vicuna 7B → GPT-4 does not show much improvement over Vicuna 7B. This indicates that planning deficiency could result in an unproductive trajectory, poses significant challenges for reasoning during the final step. Moreover, GPT-4 → Vicuna 7B shows significant regression over GPT-4, emphasizing the importance of reasoning in addition to a strong planning capability. Therefore, it is crucial for both planning and reasoning abilities to be strong and work in synergy to achieve optimal performance.

## 5 BEYOND THE WALL: TOWARDS OPEN-SOURCE MODEL ENHANCEMENT

### 5.1 BEHAVIOR CLONING

We used Behavior Cloning (BC) to distill the planning and reasoning capabilities exhibited by the stronger models into the smaller Vicuna models. We first collected game demonstrations from the GPT-3.5-turbo model over 950 training entities from *Things* and 68 names from *Celebrities* [3]. We then fine-tuned the Vicuna models using these collected demonstrations. The experimental settings are provided in the Appendix G.

**RQ3: Can smaller open-source models benefit from imitating larger close-source models?** We first performed fine-tuning on the Vicuna 7B model using a total of 1,018 training instances from both datasets. The results are presented in Table 1 as the **V-FT 7B (All)** model. This yielded a more than 70% improvement in both datasets, indicating that weaker models can leverage the knowledge from stronger models to make more precise predictions on unseen entities or names.

**RQ4: Is the transfer of knowledge generalizable across various tasks?** We wanted to understand whether this improvement stems from superficially imitating the demonstration behavior, such as copying the opening moves from a stronger model, or if it is a result of actually learning reasoning and planning strategies from the stronger model. To investigate this, we conducted two cross-domain evaluations by training two additional models on each domain separately, as presented in Table 1 as **V-FT 7B (Things)** and **V-FT 7B (Celebs)**. On evaluating these models we found that each model achieved similar performance gains on its respective domains. Interestingly, when evaluating on *Celebrities*, **V-FT 7B (Things)** exhibited nearly identical improvement compared to the **V-FT 7B (All)** model, despite the differences in the optimal opening moves of the two scenarios. A comparison of the gameplay between Vicuna 7B and the **V-FT 7B (Things)** on *Celebrities* is provided in Appendix H. We observed that **V-FT 7B (Things)** seemed to be able to transfer certain planning strategies learned from *Things* to *Celebrities*, such as avoiding early enumeration and irrelevant question, and asking high-level questions in the early game. On the other hand, the **V-FT 7B (Celebs)** failed to generalize to *Things*, evidenced by a drop in performance. Presumably, this is due to the fact that the *Celebrities* dataset is much smaller and less diverse than *Things*.

**RQ5: Should models learn from successful experiences exclusively or from all experiences, when learning from teacher's demonstration?** To answer this, we selected a total of 256 game runs of the entities or names where the teacher model finally won the game from both datasets. This subset was then used to train the Vicuna 7B model. The results, referred to as **V-FT 7B (Suc.)** in Table 1, show that imitating from successful experiences outperforms imitating from all experiences (**V-FT 7B (All)**). It even outperformed the teacher model (GPT-3.5) on *Things* , indicating the potential of self-enhancing performance through *Rejection Sampling* (Touvron et al., 2023) and *re-fine-tuning* with top-ranked game episodes, or through *Reinforcement Learning*.

**RQ6: Does the model size matter?** We further compared the performance improvement achieved through finetuning for both Vicuna 7B and 13B models. The model finetuned on Vicuna 13B, **V-FT 13B (Suc.)**, achieved higher score than **V-FT 7B (Suc.)**. However, the improvement was marginal and it under-performed the Vicuna 13B on the *Celebrities* dataset, which constitutes a smaller proportion of the data mix during the finetuning stage. This suggests that model size might not be the determining factor for performance gain, and smaller models could benefit more from BC fine-tuning.

### 5.2 REINFORCEMENT LEARNING FROM GAME-PLAY (RLGP)

We conducted further experiments to investigate whether the performance could be enhanced through learning solely from the model's own experience rather than imitating a stronger model. Drawing inspiration from Reinforcement Learning from Human Feedback (RLHF), we employed Proximal Policy Optimization (PPO) (Schulman et al., 2017) to train the model by playing with with the judge, $\mathcal{J}$, a technique we refer to as RLGP. We made modifications to the trlX repository (Castricato et al., 2023) to facilitate RLGP training. During training, we assigned the reward defined in Eq. (1), to the final turn of each rollout. Additionally, we assigned an intermediate reward to turns that received a "Yes" response from $\mathcal{J}$. We trained the Vicuna 7B and 13B models on a dataset of 209 entities from the *Things* domain. Further experimental details can be found in Appendix I.

---

[3]We opted not to use GPT-4 for generating these demonstrations due to cost and efficiency considerations.

The results in Table 1 present the performance of the RL-trained models, denoted as **V-RLGP**. **V-RLGP** models exhibit improvement compared to the vanilla Vicuna models when tested on the in-domain dataset *Things*. On the out-domain dataset *Celebrities* where the BC-trained models deteriorate, interestingly, **V-RLGP** models achieves some improvement. Notably, the difference in performance between the 7B and 13B RL-trained models is substantial. **V-RLGP 13B** seems to unlock the potential of the Vicuna model, outperforming the **V-FT 13B** model, matching the performance of the runner-up Claude-2. On the other hand, **V-RLGP 7B** fails to improve much. We hypothesize that this discrepancy could be attributed to differences in exploration efficiency between the two models, as it may take the Vicuna 7B model much more trials to generate positive rollouts.

### 5.3 BREAKDOWN ANALYSIS: DO THE MODELS AGREE ON SUCCESSFUL PREDICTIONS?

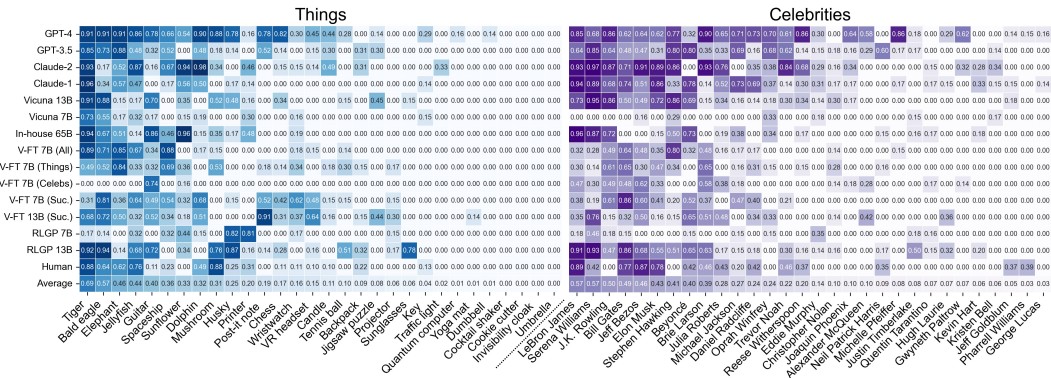

Figure 2: A breakdown of the score of each model on the evaluated items, with the x-axis representing the order of difficulty ranging from easy to difficult. Scores are averaged over 5 repetitions.

We present a comprehensive breakdown analysis of each model's performance on all the evaluated items in Figure 2. The difficulty level varies across different items. Within the evaluated items of *Things*, four items (Cocktail shaker, Cookie cutter, Invisibility cloak, and Umbrella) consistently defy successful predictions by all models. On the other hand, *Celebrities* exhibits a more scattered pattern, with each celebrity being correctly predicted by at least one model. There are correlations between the entities or names that each model can correctly identify, but different models exhibit their own strengths on different subsets. For instance, the **V-FT 7B (celebs)** model can only accurately predict "Guitar", whereas the stronger model Claude-2 consistently fails to do so. We also provide some case studies including why GPT-4 consistently fails on "Yoga mat" in Appendix K.

We observed that RLGP models tend to strengthen the performance on items that vanilla models occasionally succeed in, thereby improving their success rate on these specific items. However, RLGP models do not effectively facilitate learning about new items. Conversely, BC fine-tuning appears to assist the model in achieving success on new items. Interestingly, the model that undergoes BC fine-tuning displays different strengths compared to both the initial checkpoint and the expert whose demonstration it mimics. For example, the **V-FT 13B (Suc.)** model achieves high accuracy in identifying Post-it Note and VR headset, whereas neither the Vicuna 13B nor the GPT-3.5 performs as well in this regard.

## 6 CONCLUSION

Motivated by a need to develop agents capable of effectively addressing ambiguous user intents, we introduce a testbed for evaluating LLM's strategic planning and deductive reasoning abilities in asking entity-deducing questions. Our findings indicate that SOTA LLMs are able to maintain an intrinsic taxonomic representation of knowledge entities to a certain extent. We further show that this capability can be enhanced through Behavior Cloning or Reinforcement Learning, revealing great potential for further advancements. In future research, we intend to investigate whether the implementation of CoT prompting can further enhance the model's performance in related tasks.

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
