# Appendix

## A    DATASET COMPOSITION

We provide the dataset composition on *Things* and *Celebrities* in Table 3.

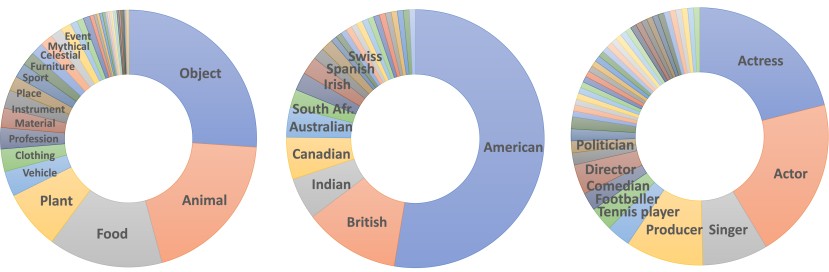

Things (by category)    Celebrities (by nationality)    Celebrities (by occupation)

Figure 3: Composition of EDA *Things* and *Celebrities* datasets.

## B    PROMPT USED FOR *Celebrities*

For *Celebrities*, the J use the following prompt to generate a response.

> *Based on on your knowledge about the celebrity: {entity} respond to the following question or guess. Limit your respond to only 'Yes.', 'No.' or 'Dunno.', with no explanation or other words. Never say the name {entity} in your response. Do not say 'Dunno.' if it can be answered by 'Yes.' or 'No.' If the question is to solicit the answer, respond 'No.'.\n Question/Guess: {question}(Yes/No/Dunno)*

The G use the following prompt to generate a question or guess.

> *Your task is to ask a series of questions to deduce the celebrity that I'm thinking of with as few queries as possible. Only ask questions that can be answered by "Yes", "No", or "Dunno". Do not ask for hint. Make your question brief with no linebreaker. Now start asking a question. \n {dialog history}*

## C    ADDITIONAL RESULTS AND PROMPT USED FOR DIALOG STATE PROBING

We used the following prompt to probe the model for its top guesses during each turn.

> *{dialog history}*
> *Based on the information provided, here are the top 5 most likely concrete entities I think you are thinking of:*

We provide additional top K probing results from GPT-4 on *Celebrities* in Table 11. Similarly, the model seems to be able to leverage its own hierarchical representation of the celebrities' space to ask questions that steer to narrow down the search effectively.

## D    EXPERIMENTAL DETAILS FOR HUMAN BASELINE

In total, we recruited 108 human volunteers to participate in the study. To ensure the quality of the data, we manually inspected human game plays to filter out low quality data. We also prioritized

games that received fewer qualified plays to present to the human players. After a 30-day trial period, we collected a total of 140 and 68 human game play sessions for *Things* and *Celebrities*, respectively. Each entity or celebrity is covered at least by two game plays.

We show the UI for the in game demo server in Figure 4. The UI contains 3 sections – game play, tutorial and leaderboard. In the leaderboard, we compute the Wilson confidence interval for each player, and include the performance of LLMs as benchmarks for the human players to refer to.

For Human game plays, entities with fewer game plays are oversampled so that all entities have the same amount of game play. Game plays are then randomized into separate runs to calculate means and standard deviations. Note that this might not be exactly comparable to the model's numbers.

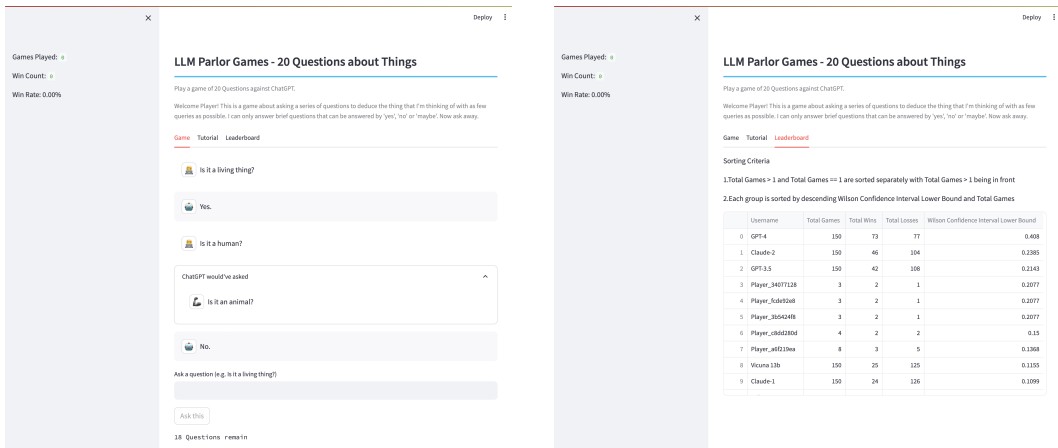

Figure 4: Game play UI interface for collecting human baseline. On the left, human players are given prompt instructions equivalent to LLM guessers. An optional retrospection UI can be toggled to display what ChatGPT would've chosen to ask in the last turn. On the right, a leaderboard with Human and LLM player performance is shown.

# E    COMPARISON OF GPT-4 VS HUMAN PLAYER

We provide additional examples in Table 6 (*Things*) and Table 7 (*Celebrities*) for GPT-4's generation and compare them with the human players.

## E.1    *Things* DATASET

In the first example "Guitar", we observed that both GPT-4 and the human player efficiently converged on a music-related object within approximately 10 turns. However, the human player deviated from the correct path by thinking about an electronic device, whereas GPT-4 promptly identified the item as a musical instrument. Notwithstanding this, the human player's capacity for back-tracking their steps was evident when they recognized the erroneous direction and ultimately win the game. In this game, the human player retraced the error and asking a reconfirming question "Is it some kind of music player?".

We then ask the question if the GPT-4 model also has this back-tracking ability. The answer is yes. In Table 6 (Guitar, Human + GPT-4), we ask GPT-4 to complete the game based on partial gameplay provided by human players until they recognize their mistake (up to "G: Can it display images? J: No."). GPT-4 promptly comprehends the game scenario and identifies that it has taken the incorrect route. Subsequently, it formulates the appropriate question that encompasses the overlooked category, thereby salvaging the game. We have also observed this similar capability in several other game sessions from GPT-4.

In the second game "Bald eagle", human player has a luck guess after identifying the entity is a bird. GPT-4, on the contrary, still roll out 3 more turns to gather more information before making an informative guess.

In the third example "Jigsaw Puzzle", GPT-4 interleaves questions regarding size, purpose and location and combine multiple aspects of information to make a guess. Prior to making the final guess, GPT-4 demonstrates caution by reconfirming if the object is a physical toy, after receiving a "No" with "Is it a board game?". On the contrary, the human player seems clueless and may ask repetitive or irrelevant questions.

### E.2 *Celebrities* DATASET

We present a comparative analysis between GPT-4 and human players on *Celebrities* in Table 7. Our findings reveal that GPT-4 demonstrates a propensity for asking questions that necessitate domain-specific background knowledge (as highlighted in Table 7). Such knowledge could be *tacit knowledge* between LLMs, which human players may lack. With its extensive information repository, GPT-4's inquiries can delve into highly specific topics, sometimes requiring professional expertise that human judges may not possess. This characteristic confers an additional strength and advantage to GPT-4 during gameplay. Conversely, human players tend to rely more on direct guesses, which may yield fortunate outcomes in certain scenarios, but limits their overall effectiveness when compared to GPT-4.

## F  GAME RUNS AND FAILURE MODES ON *Celebrities* AND *Things*

We provide additional gameplay sessions on *Celebrities*, as shown in Table 8. Similar findings can be observed as they are in the experiment section. We observed that stronger models display proficient gameplay by accurately identifying the relevant question that separates the current space, while weaker models encounter difficulties in planning and reasoning.

We also presents several other failure modes observed with the Vicuna 7b model in Table 9. These failure modes illustrate the repetitive behavior and generation of artifacts. When a less robust model is drawn towards an absorbing state of repetitive pattern, it becomes increasingly challenging for them to escape from it.

## G  EXPERIMENTAL SETTINGS FOR BAHAVIOR CLONING

We used 4x A100 Nvidia GPU for training. The learning rate was configured to be 2e-5, while the batch size was set to 32. We utilized the BF16 method for training and consistently employed deepspeed stage 2 across all our experiments. As part of the training process, we masked tokens from the turns of the J within the loss function. In all of our experiments, we continued training the model until the validation loss ceased to decrease any further.

## H  GENERALIZATION ABILITY OF VICUNA 7B FINETUNED ON *Things* TO *Celebrities*

A comparison of the gameplay between Vicuna 7B and the **V-FT 7B (Things)** is provided in Table 10. We observe that Vicuna 7B model may be more inclined to ask irrelevant questions, while the V-FT 7B (things) ask questions directly to the point. In terms of game strategy, the V-FT 7B (things) is more generative effective, indicating the planning ability obtained from imitation might be able to generalize to out of domain scenario.

## I  EXPERIMENTAL SETTINGS FOR REINFORCEMENT LEARNING FROM GAME-PLAY (RLGP)

We adhered to the default settings of the TRLX library for our experiments, but made modifications to certain hyperparameters. Specifically, we set the rollout generation temperature to 0.8 and disabled the topK or topP sampling. The value function coefficient for the value hydra head was set to 0.05. For the discounted reward, we set $\gamma$ to 1 and $\lambda$ to 0.97. The RL models were trained using four Nvidia A100 cards, with a total of 32 rollouts for each training entity. We trained the Vicuna 7B and 13B models on a dataset of 209 entities from the *Things* domain, which at least one of the off-the-shelf

LLMs had won at least once. This is to ensure that model learn from experiences where it has some chance to receive some final rewards.

The intermediate reward was determined as follows. We utilized a linear decaying function to gradually decrease the intermediate reward from 2 to 0, based on the following equation:

$$R_{\text{intermediate}} = \max(0.2 - 0.025 \cdot \textbf{\#Turns}, 0) \tag{2}$$

The rationale behind this choice is that achieving more "yes" responses in the earlier turns is considered a positive signal. We observed that this approach improved the performance of the RLGP model. If the initial intermediate reward is too high (*e.g.,* 0.5), the model tends to generate trivial questions like "is it an entity" more frequently. Our choice is based the empirical results obtained from the development set. All RLGP models were trained for 600 iterations, ensuring that the model does not deviate significantly from the initial Vicuna model.

## J  FULL ABLATION RESULTS ON RQ2, PLANNING VS REASONING

We provide the full results in Table 5 for this ablation.

| | Things | | | | Celebrities | | | |
|---|---|---|---|---|---|---|---|---|
| | #Turns (↓) | Success (↑) | #Yes | Score (↑) | #Turns (↓) | Success (↑) | #Yes | Score (↑) |
| GPT-4 | **16.9±0.2** | **0.49±0.06** | 6.0±0.2 | **0.40±0.05** | 16.5±0.5 | **0.59±0.04** | 7.3±0.1 | **0.48±0.03** |
| GPT-4 → Vicuna 7B | 16.9±0.2 | 0.14±0.04 | 6.1±0.2 | 0.12±0.03 | 16.5±0.5 | 0.23±0.03 | 7.3±0.1 | 0.19±0.02 |
| Vicuna 7B | 19.1±0.3 | 0.12±0.05 | 5.8±0.5 | 0.10±0.04 | 19.7±0.3 | 0.05±0.04 | 6.2±0.7 | 0.04±0.03 |
| Vicuna 7B → GPT-4 | 19.1±0.3 | 0.14±0.04 | 5.7±0.2 | 0.11±0.03 | 19.7±0.3 | 0.09±0.04 | 6.1±0.6 | 0.06±0.03 |

Table 5: Planning and reasoning ability comparison between Vicuna 7B and GPT-4. GPT-4 → Vicuna 7B uses GPT-4 to play all except the last turn, swapping in Vicuna 7B in the end to perform reasoning, where as Vicuna 7B → GPT-4 does the opposite.

## K  GPT-4 FAILURE CASES

We notice that the strongest model GPT-4 consistently fail on the entity "Yoga Mat", while some other models can adeptly deduct it. We investigated this issue and it turns out that the GPT-4 model will always ask a misleading question "Is it made of plastic" in the 5 tested runs, leading to self-misinterpretation. See examples in Table 12. GPT-4 seems to possess a stereotype of plastic items and excludes "Yoga mat" after knowing this item is made of plastic. The effect of this question is more evident in Table 13. After confirming that the entity is made of plastic, the model proceeds to focus on plastic only items in its top guesses for all following turns. This observation suggests that even the most robust model, which may demonstrate backtracking when encountering a wrong path, can still become trapped in erroneous assumptions.

| Item: Guitar | | |
|---|---|---|
| GPT-4 | Human player | Human + GPT-4 |
| G: Is it a living entity? *J: No.*
G: Is it man-made? *J: No.*
G: Can it be held in one hand? *J: Yes.*
G: Is it electronic? *J: No.*
G: Is it used for writing or drawing? *J: No.*
G: Is it made of metal? *J: No.*
G: Is it used in cooking or eating? *J: No.*
G: Is it used for entertainment or leisure? *J: Yes.*
G: Is it associated with a specific sport? *J: No.*
G: Is it used in board games? *J: No.*
G: Is it associated with music? *J: Yes.*
G: Is it a musical instrument? *J: Yes.*
G: Is it a stringed instrument? *J: Yes.*
G: Is it commonly used in a symphony orchestra? *J: No.*
G: Is it a guitar? *J: Bingo!* | G: Is it a living thing? *J: No.*
G: Is it a man-made object? *J: Yes.*
G: Is it commonly used indoors? *J: Yes.*
G: Is it larger than a toaster? *J: Yes.*
G: Is it used for recreational purposes? *J: Yes.*
G: Does it require electricity? *J: Maybe.*
G: Does it have a screen? *J: No.*
G: Does it contain metal? *J: Yes.*
G: Does it contain glass? *J: No.*
G: Does it make sound? *J: Yes.*
G: Is it used to play music? *J: Yes.*
G: Is it a speaker? *J: No.*
G: Is it a radio? *J: No.*
G: Does it have a wireless connection? *J: No.*
G: Can it display images? *J: No.*
G: **Is it some kind of music player?** *J: No.*
G: Is it a musical instrument? *J: Yes.*
G: Is it a guitar? *J: Bingo!* | G: Is it a living thing? *J: No.*
G: Is it a man-made object? *J: Yes.*
G: Is it commonly used indoors? *J: Yes.*
G: Is it larger than a toaster? *J: Yes.*
G: Is it used for recreational purposes? *J: Yes.*
G: Does it require electricity? *J: Maybe.*
G: Does it have a screen? *J: No.*
G: Does it contain metal? *J: Yes.*
G: Does it contain glass? *J: No.*
G: Does it make sound? *J: Yes.*
G: Is it used to play music? *J: Yes.*
G: Is it a speaker? *J: No.*
G: Is it a radio? *J: No.*
G: Does it have a wireless connection? *J: No.*
G: Can it display images? *J: No.*
GPT: Is it a musical instrument? *J: Yes.*
GPT: Is it a string instrument? *J: Yes.*
GPT: Is it a guitar? *J: Bingo!* |

| Item: Bald eagle | |
|---|---|
| GPT-4 | Human player |
| G: Is it a living thing? *J: Yes.*
G: Is it an animal? *J: Yes.*
G: Is it a domestic animal? *J: No.*
G: Is it a mammal? *J: No.*
G: Is it a reptile? *J: No.*
G: Is it a bird? *J: Yes.*
G: Is it a bird of prey? *J: Yes.*
G: Does it have a predominantly white color? *J: Yes.*
G: Is it found in the Polar regions? *J: No.*
G: Is it a bald eagle? *J: Bingo!* | G: Is it alive? *J: Yes.*
G: Does it live on land? *J: Yes.*
G: Is it a plant? *J: No.*
G: Is it human? *J: No.*
G: Does it live in the Americas? *J: Yes.*
G: Does it live in North America? *J: Yes.*
G: Is it a predator? *J: Yes.*
G: Is it a bear? *J: No.*
G: Is it a bird? *J: Yes.*
G: Is it a bald eagle? *J: Bingo!* |

| Item: Jigsaw Puzzle | |
|---|---|
| GPT-4 | Human player |
| G: Is it a living entity? *J: No.*
G: Is it man-made? *J: Yes.*
G: Is it used for communication? *J: No.*
G: Is it larger than a microwave? *J: Maybe.*
G: Is it used for transportation? *J: No.*
G: Is it made of metal? *J: No.*
G: Is it used in homes? *J: Yes.*
G: Is it used for sitting? *J: No.*
G: Is it used for storage? *J: No.*
G: Is it a decorative item? *J: Maybe.*
G: Is it used in the kitchen? *J: No.*
G: Is it used in the living room? *J: Maybe.*
G: Is it used for entertainment purposes? *J: Yes.*
G: Is it a musical instrument? *J: No.*
G: Is it a television? *J: No.*
G: Is it a game or a toy? *J: Yes.*
G: Is it a board game? *J: No.*
G: Is it a video game console? *J: No.*
G: Is it a physical toy? *J: Yes. You must guess now, what's it?*
G: Is it a puzzle? Is it right? *J: Yes.* | G: What is the meaning of life? *J: No.*
G: What is a good thing? *J: Maybe.*
G: Is it human? *J: No.*
G: Is it an object? *J: Yes.*
G: Is it a tool? *J: Yes.*
G: Is it a hammer? *J: No.*
G: Is it a kitchen tool? *J: No.*
G: What is it? *J: Maybe.*
G: Does it cut? *J: No.*
G: Is it for food? *J: No.*
G: Is it for cars? *J: No.*
G: Is it big? *J: Maybe.*
G: Can you eat it? *J: No.*
G: Is it red? *J: No.*
G: Is it metal? *J: No.*
G: Is it used for communication? *J: No.*
G: Is it used for storing? *J: No.*
G: Is it used for helping humans? *J: Maybe.*
G: Is it used for taking pictures? *J: No.*
G: Alarm clock *J: No.* |

Table 6: The gameplay scenario generated using GPT-4 model vs the gameplay scenario by human players, on *Things*.

| Celebrity: Reese Witherspoon | |
|---|---|
| GPT-4 | Human player |
| G: Is the celebrity you're thinking of alive? *J: Yes.*
G: Is this celebrity a man? *J: No.*
G: Is this celebrity an actress? *J: Yes.*
G: Is she American? *J: Yes.*
G: Has she won an Oscar? *J: Yes.*
G: Is she over 50 years old? *J: No.*
G: Has she starred in a superhero movie? *J: No.*
G: Is she blonde? *J: Yes.*
G: **Has she appeared in a film directed by Martin** **Scorsese?** *J: No.*
G: Has she won her Oscar within the last 10 years? *J: No.*
G: Is she known for romantic comedy films? *J: Yes.*
G: Is it Reese Witherspoon? *J: Bingo!* | G: Male? *J: No.*
G: White? *J: Yes.*
G: American *J: Yes.*
G: American *J: Yes.*
G: Actor *J: Yes.*
G: Alive *J: Yes.*
G: Action movies *J: No.*
G: Romantic comedies *J: Yes.*
G: Sandra bullock *J: No.*
G: Over 50 years old *J: No.*
G: Emma stone *J: Dunno.*
G: Is the person female? *J: Yes.*
G: Meg Ryan *J: No.*
G: Amy Schumer *J: No.*
G: Kate Hudson *J: No.*
G: Julia Roberts *J: No.*
G: Cameron Diaz *J: No.*
G: Reese Witherspoon *J: Bingo!* |
| Celebrity: Trevor Noah | |
| GPT-4 | Human player |
| G: Is the celebrity you're thinking of alive? *J: Yes.*
G: Is the celebrity an actor? *J: Yes.*
G: Is the celebrity American? *J: No.*
G: Is the celebrity male? *J: Yes.*
G: Has the celebrity won an Oscar? *J: No.*
G: Is the celebrity known for roles in action films? *J: No.*
G: Is the celebrity from the United Kingdom? *J: No.*
G: Is the celebrity from Australia? *J: No.*
G: Is the celebrity from Asia? *J: No.*
G: Is the celebrity from Europe? *J: No.*
G: Is the celebrity from Canada? *J: No.*
G: Is the celebrity from South America? *J: No.*
G: **Is the celebrity from Africa?** *J: Yes.*
G: Is the celebrity known for roles in comedic films? *J: Yes.*
G: Is the celebrity Trevor Noah? *J: Bingo!* | G: Is the person male? *J: Yes.*
G: Is the person on TV? *J: Yes.*
G: Does the person host a late night show? *J: Yes.*
G: Is the show factual? *J: Dunno.*
G: Is the person Stephen Colbert *J: No.*
G: Is the person a news anchor? *J: No.*
G: Is the person a comedian? *J: Yes.*
G: Is the person John Oliver? *J: No.*
G: Is the person Trevor Noah? *J: Bingo!* |
| G: Is your celebrity alive? *J: Yes.*
G: Is your celebrity an actor? *J: Yes.*
G: Is your celebrity female? *J: No.*
G: Has your celebrity won an Oscar? *J: No.*
G: Is your celebrity American? *J: No.*
G: Is your celebrity British? *J: No.*
G: Is your celebrity from Australia? *J: No.*
G: Is your celebrity from Asia? *J: No.*
G: Is your celebrity from Europe? *J: No.*
G: Is your celebrity from Canada? *J: No.*
G: Is your celebrity from Africa? *J: Yes.*
G: Has your celebrity starred in a superhero movie? *J: No.*
G: Is your celebrity over the age of 50? *J: No.*
G: **Has your celebrity acted in a film that was nominated** **for an Academy Award?** *J: Yes.*
G: Is your celebrity from South Africa? *J: Yes.*
G: Is your celebrity Trevor Noah? *J: Bingo!* | G: Is your celebrity a movie star? *J: Yes.*
G: Is the celebrity female? *J: No.*
G: Is the celebrity over 50 years old? *J: No.*
G: Is the celebrity over 30 years old? *J: Yes.*
G: Has this celebrity been in a Marvel movie? *J: No.*
G: Is this celebrity known for comedies? *J: Yes.*
G: Is this celebrity african american? *J: No.*
G: Has this celebrity also been in a TV show? *J: Yes.*
G: Has this celebrity been in a movie grossing more than a billion dollars? *J: No.*
G: Is this celebrity still making hit movies? *J: No.*
G: Was this celebrity famous in the 90s? *J: No.*
G: Was this celebrity famous in the 2000s? *J: Yes.*
G: Has this celebrity worked with Judd Apatow? *J: No.*
G: Was this celebrity in one of the Hangover movies? *J: No.*
G: Is this celebrity American? *J: Yes.*
G: Was this person on network TV? *J: Yes.*
G: Does this celebrity have a famous spouse? *J: No.*
G: Is this celebrity over 40 years old? *J: No.*
G: Is it Zac Efron? *J: No.*
G: Is it Zachary Levi? *J: No.* |

Table 7: The gameplay scenario generated using GPT-4 model vs the gameplay scenario by human players, on *Things*. The questions need in-domain knowledge is **highlighted**.

| Successful game | Planning issue (Early Enumeration) |
|---|---|
| G: Is the celebrity alive? *J: Yes.*
G: Is the celebrity a male? *J: No.*
G: Is she an actress? *J: Yes.*
G: Is she American? *J: Yes.*
G: Has she won an Academy Award? *J: Yes.*
G: Is she above 50 years old? *J: No.*
G: Has she starred in a superhero movie? *J: Yes.*
G: Is she in the Marvel Cinematic Universe? *J: Yes.*
G: Did she play a character that has their own standalone film in the Marvel Cinematic Universe? *J: Yes.*
G: Has she played a human with supernatural powers? *J: No.*
G: Did she play the character of Black Widow? *J: No.*
G: Did she play Captain Marvel? *J: No.*
G: Did she play the character of Wasp in the Ant-Man series? *J: Yes.*
G: Is the celebrity Evangeline Lilly? *J: No.*
G: Did she play the character of Gamora in the Guardians of the Galaxy series? *J: No.*
G: Did she play the character of Pepper Potts in the Iron Man series? *J: Yes.*
G: Is the celebrity Gwyneth Paltrow? *J: Bingo!* | G: Is the celebrity a person who has been in the public eye for at least 10 years? *J: Yes.*
G: Is the celebrity primarily known for acting? *J: Yes.*
G: Has the celebrity won an Academy Award? *J: No.*
G: Has the celebrity been in any Marvel Cinematic Universe films? *J: Yes.*
G: Is the celebrity primarily known for their work in the **Marvel Cinematic Universe**? *J: No.*
G: Has the celebrity been in any **DC Extended Universe films**? *J: No.*
G: Has the celebrity been in any **Star Wars films**? *J: No.*
G: Is the celebrity primarily known for their work in television? *J: No.*
G: Has the celebrity been in any reality TV shows? *J: No.*
... |

| Planning issue (Redundancy) | Reasoning issue (Inconsistency) |
|---|---|
| G: Is the celebrity a musician? *J: No.*
G: Is the celebrity primarily known for their acting career? *J: Yes.*
G: Is the celebrity a former child actor? *J: No.*
G: Is the celebrity primarily known for their work in film? *J: Yes.*
G: Is the celebrity mostly known for their work in the past decade? *J: Yes.*
G: Is the celebrity a **British national**? *J: No.*
G: Is the celebrity primarily known for their work in American cinema? *J: Yes.*
G: Is the celebrity a former child star who later became a Hollywood movie star? *J: Yes.*
G: Is the celebrity Emma Watson? *J: No.*
G: Is the celebrity a **male actor**? *J: No.*
G: Is the celebrity a **British female actor**? *J: No.*
... | G: Is the celebrity **male**? *J: Yes.*
G: Has the celebrity won an Academy Award? *J: No.*
G: Is the celebrity primarily known for their work in film? *J: No.*
G: Is the celebrity primarily known for their work in television? *J: No.*
G: Is the celebrity known for their work in music? *J: No.*
G: Does the celebrity have a significant presence on social media? *J: Yes.*
G: Is the celebrity a comedian? *J: No.*
G: Is the celebrity a singer? *J: No.*
G: Is the celebrity a dancer? *J: No.*
...
G: Is the celebrity a political figure? *J: No. You must guess now*
G: The celebrity is **Ellen DeGeneres**. Is it right? *J: No.* |

Table 8: Demonstration of failure modes of the gameplay with hidden celebrity names. The successful gameplay scenario is generated using GPT-4 model as G. The remaining gameplay scenarios are generated using vicuna-7b model as G.

| Artifact: Chain Reaction | Artifact: Self-answering |
|---|---|
| G: Is the entity a living thing? *J: Yes.*
G: Is the living thing an animal? *J: Yes.*
G: Is the animal a mammal? *J: Yes.*
G: Is the mammal a warm-blooded? *J: Yes.*
G: Is the warm-blooded mammal a primate? *J: No.*
G: Is the primate a marmoset? *J: No.*
G: Is the marmoset a rodent? *J: No.*
G: Is the rodent a squirrel? *J: No.*
G: Is the squirrel a rabbit? *J: No.* | G: Is the entity you are thinking of a person? Yes *J: No.*
G: Is the entity you are thinking of an animal? Yes *J: Yes.*
G: Is the animal you are thinking of a mammal? Yes *J: No.*
G: Is the animal you are thinking of a bird? Yes *J: No.*
G: Is the animal you are thinking of a reptile? Maybe *J: No.*
G: Is the animal you are thinking of an amphibian? Maybe *J: No.*
G: Is the animal you are thinking of a fish? Yes *J: No.*
G: Is the animal you are thinking of a crustacean? Maybe *J: No.*
G: Is the animal you are thinking of an arachnid? Maybe *J: No.*
G: Is the animal you are thinking of an insect? Yes *J: No.*
G: Is the animal you are thinking of a mollusk? No *J: No.*
... |

Table 9: Demonstration of failure modes from Vicuna-7b model where the model was trapped by repetitive or problematic patterns.

| Celebrity: LeBron James | |
|---|---|
| Vicuna 7B | V-FT 7B (Things) |
| G: Was this celebrity a singer? *J: No.*
G: Was this celebrity a musician? *J: No.*
G: Was this celebrity an actor? *J: No.*
G: Was this celebrity a comedian? *J: No.*
G: Was this celebrity a public figure? *J: Yes.*
G: Was this public figure a politician? *J: No.*
G: Was this public figure a sports person? *J: Yes.*
G: Was this sports person a professional athlete? *J: Yes.*
G: Was this professional athlete a tennis player? *J: No.*
G: Was this professional athlete a boxer? *J: No.*
G: Was this professional athlete a golfer? *J: No.*
G: Was this professional athlete a football player? *J: No.*
G: Was this professional athlete a basketball player? *J: Yes.*
G: Was this basketball player a member of the NBA? *J: Yes.*
G: Was this basketball player active during the 1990s? *J: No.*
G: Was this basketball player active during the 2000s? *J: Yes.*
G: Was this basketball player known for his clutch performances? *J: Yes.*
G: Can you name the basketball player? *J: No.*
G: I'm sorry, I'll have to ask for a hint. Could it be someone who played for the Los Angeles Lakers during the 2000s? *J: No. You must guess now, what's it?*
G: Could it be someone who played for the Los Angeles Clippers during the 2000s? Is it right? *J: No.* | G: Is the celebrity you're thinking of a male? *J: Yes.*
G: Is the celebrity you're thinking of a singer? *J: No.*
G: Is the celebrity you're thinking of an actor? *J: No.*
G: Is the celebrity you're thinking of a sports player? *J: Yes.*
G: Is the celebrity you're thinking of a male sports player? *J: Yes.*
G: Is the celebrity you're thinking of an Olympic athlete? *J: Yes.*
G: Is the celebrity you're thinking of a swimmer? *J: No.*
G: Is the celebrity you're thinking of a track and field athlete? *J: No.*
G: Is the celebrity you're thinking of a gymnast? *J: No.*
G: Is the celebrity you're thinking of a boxer? *J: No.*
G: Is the celebrity you're thinking of a soccer player? *J: No.*
G: Is the celebrity you're thinking of a basketball player? *J: Yes.*
G: Is the celebrity you're thinking of a former NBA player? *J: Yes.*
G: Was the celebrity you're thinking of a dominant scorer? *J: Yes.*
G: Did the celebrity you're thinking of have a reputation for being a high-flyer? *J: Yes.*
G: Is the celebrity you're thinking of known for their acrobatic dunks? *J: Yes.*
G: Is the celebrity you're thinking of Michael Jordan? *J: No.*
G: Is the celebrity you're thinking of Kobe Bryant? *J: No.*
G: Is the celebrity you're thinking of LeBron James? *J: Bingo!* |
| Celebrity: Brie Larson | |
| Vicuna 7B | V-FT 7B (Things) |
| G: Is the celebrity you're thinking of a male? *J: No.*
G: Does the celebrity you're thinking of have a sister? *J: Yes.*
G: Is the sister of your celebrity the same gender as the celebrity? *J: Yes.*
G: Is the celebrity you're thinking of a musician? *J: No.*
G: Is the celebrity you're thinking of a member of a pop group? *J: No.*
G: The celebrity is not a musician, not a member of a pop group, and not a male. Do you know their name? *J: No.*
G: You have provided the name of the celebrity. Is it a living person? *J: Yes.*
G: Your celebrity is a living person who is not a musician, not a member of a pop group, and not a male. Can you provide their profession? *J: Yes.*
... | G: Is the celebrity male? *J: No.*
G: Is the celebrity a singer? *J: No.*
G: Is the celebrity an actress? *J: Yes.*
G: Has the celebrity won an Academy Award? *J: Yes.*
G: Has the celebrity won the Academy Award for Best Actress? *J: Yes.*
G: Has the celebrity won multiple Academy Awards for acting? *J: No.*
G: Did the celebrity win the Academy Award for Best Actress for her role in a biographical film? *J: Yes.*
G: Does the celebrity have blonde hair? *J: No.*
G: Has the celebrity played a superhero character in a movie? *J: Yes.*
G: Is the celebrity over the age of 40? *J: No.*
G: Has the celebrity won Academy Awards for portraying real-life individuals on screen? *J: Yes.*
G: Is the celebrity Charlize Theron? *J: No.*
G: Is the celebrity Cate Blanchett? *J: No.*
G: Is the celebrity Jennifer Lawrence? *J: No.*
G: Is the celebrity Brie Larson? *J: Bingo!* |

Table 10: Comparison of gameplay scenario on *Celebrities* between Vicuna 7B model vs **V-FT 7B (Things)**.

| Celebrity to be deducted: LeBron James | | | | |
|---|---|---|---|---|
| Current top 4 guesses before generating question (Ranked) | | | | Generated question |
| Leonardo DiCaprio | ~~Jennifer Aniston~~ | ~~Will Smith~~ | ~~Angelina Jolie~~ | G: Is the celebrity male? *J: Yes.* |
| ~~Brad Pitt~~ | ~~Tom Cruise~~ | ~~Leonardo DiCaprio~~ | ~~Dwayne Johnson~~ | G: Is he an actor? *J: No.* |
| ~~Justin Bieber~~ | ~~Drake~~ | Elon Musk | Cristiano Ronaldo | G: Is he a musician? *J: No.* |
| ~~Donald Trump~~ | Elon Musk | Cristiano Ronaldo | ~~Richard Branson~~ | G: Is he involved in sports? *J: Yes.* |
| ~~Michael Jordan~~ | ~~Tiger Woods~~ | *LeBron James* | Cristiano Ronaldo | G: Is he a retired athlete? *J: No.* |
| Cristiano Ronaldo | *LeBron James* | Lionel Messi | Tom Brady | G: Does he play a team sport? *J: Yes.* |
| ~~Cristiano Ronaldo~~ | Lionel Messi | *LeBron James* | ~~Tom Brady~~ | G: Does he play American football? *J: No.* |
| *LeBron James* | ~~Lionel Messi~~ | ~~Sidney Crosby~~ | Stephen Curry | G: Does he play basketball? *J: Yes.* |
| *LeBron James* | Stephen Curry | Kevin Durant | ~~Kawhi Leonard~~ | G: Is he currently playing in the NBA? *J: Yes.* |
| *LeBron James* | ~~Stephen Curry~~ | ~~Kevin Durant~~ | ~~Giannis Antetokounmpo~~ | G: Does he play for the Los Angeles Lakers? *J: Yes.* |
| *LeBron James* | ~~Anthony Davis~~ | ~~Russell Westbrook~~ | ~~Carmelo Anthony~~ | G: Is he LeBron James? *J: Bingo!* |

Table 11: Top 4 guesses made by GPT-4 during a game of guessing LeBron James. Guesses are generated before asking the clarification questions in the same line. ~~Striked guesses~~ are eliminated after the question is answered in a retrospective perspective.

| Repetition 1 | Repetition 2 |
|---|---|
| G: Is it a living entity? *J: No.* | G: Is it a living entity? *J: No.* |
| G: Is it man-made? *J: Yes.* | G: Is it man-made? *J: Yes.* |
| G: Is it used for communication? *J: No.* | G: Is it something used daily? *J: Maybe.* |
| G: Is it smaller than a bread box? *J: Yes.* | G: Is it electronic? *J: No.* |
| G: Is it electronic? *J: No.* | G: Is it something you can hold in your hand? *J: Yes.* |
| G: Is it made of metal? *J: No.* | G: Is it used for writing or drawing? *J: No.* |
| G: Is it used in daily life? *J: Yes.* | **G: Is it made of plastic?** *J: Yes.* |
| G: Is it associated with personal hygiene? *J: Maybe.* | G: Is it used in the kitchen? *J: No.* |
| G: Is it a container of sorts? *J: No.* | G: Is it a toy? *J: No.* |
| G: Is it used in cooking or eating? *J: No.* | G: Is it used for personal care or hygiene? *J: No.* |
| G: Is it used for writing or drawing? *J: No.* | G: Is it used for storage? *J: No.* |
| G: Is it associated with clothing or fashion? *J: No.* | G: Is it a tool or instrument? *J: Yes.* |
| G: Is it made of glass? *J: No.* | G: Is it associated with a specific profession or hobby? *J: Maybe.* |
| **G: Is it made of plastic?** *J: Yes.* | G: Is it used for measuring? *J: No.* |
| G: Is it found in a bathroom? *J: No.* | G: Does it have moving parts? *J: No.* |
| G: Is it found in a bedroom? *J: Maybe.* | G: Is it used in crafting or DIY projects? *J: Maybe.* |
| G: Is it a toy? *J: No.* | G: Is it a ruler? *J: No.* |
| G: Is it used for storage? *J: No.* | G: Is it a magnifying glass? *J: No.* |
| G: Is it used for cleaning purposes? *J: No. You must guess now, what's it?* | G: Is it a stencil? *J: No. You must guess now, what's it?* |
| G: Is it a plastic comb? Is it right? *J: No.* | G: Is it a glue stick? Is it right? *J: No.* |

Table 12: GPT-4 consistently fails on "Yoga mat" after confirming the entity is made of plastic.

| Entity to be deducted: Yoga Mat | | | |
|---|---|---|---|
| Current top 3 guesses before generating question (Ranked) | | | Generated question |
| ~~computer~~ | ~~tree~~ | ~~book~~ | G: Is it a living entity? *J: No.* |
| ~~Eiffel Tower~~ | ~~Great Pyramid of Giza~~ | ~~Statue of Liberty~~ | G: Is it a living tangible? *J: Yes.* |
| book | ~~car~~ | ~~computer~~ | G: Is it a man-made? *J: Yes.* |
| ~~skyscraper~~ | book | smartphone | G: Can it be used in everyday life? *J: Yes.* |
| ~~smartphone~~ | ~~car~~ | book | G: Is it electronic? *J: No.* |
| book | ~~chair~~ | ~~table~~ | G: Is it smaller than a bread box? *J: Yes.* |
| ~~pencil~~ | book | ~~pair of glasses~~ | G: Is it made of metal? *J: No.* |
| ~~pen~~ | ~~book~~ | ~~wine glass~~ | G: Is it made out of paper? *J: No.* |
| ~~wooden spoon~~ | ~~glass cup~~ | plastic comb | **G: Is it made of plastic?** *J: Yes.* |
| ~~plastic water bottle~~ | ~~plastic comb~~ | plastic spoon | G: Is it used for personal hygiene? *J: No.* |
| ~~plastic spoon~~ | ~~plastic cup~~ | plastic toy | G: Is it used in the kitchen? *J: No.* |
| plastic pen | plastic ruler | ~~plastic toy~~ | G: Is it used for entertainment or leisure? *J: No.* |
| ~~plastic pen~~ | plastic comb | ~~plastic ruler~~ | G: Is it used in an office setting? *J: No.* |
| plastic comb | plastic spoon | plastic toy car | G: Is it related to education or learning? *J: No.* |
| ~~plastic comb~~ | ~~plastic spoon~~ | ~~plastic toy car~~ | G: Is it associated with storage or organization? *J: Maybe.* |
| plastic storage box | plastic organizer drawer | ~~plastic file folder~~ | G: Is it found in a bedroom? *J: Maybe.* |
| plastic box for storage | ~~plastic hanger~~ | ~~plastic drawer organizer~~ | G: Is it associated with clothing or fashion? *J: No.* |
| plastic box or container | plastic toy bin | ~~plastic shoe rack~~ | G: Is it typically found in a garage or outdoors? *J: No.* |
| ~~plastic storage box~~ | ~~plastic toy box~~ | ~~plastic jewelry box~~ | G: Is it used to hold or contain smaller items? *J: No.* |
| plastic cup | plastic ruler | plastic toy | G: Is it a plastic ruler? *J: No.* |

Table 13: Top 3 guesses made by GPT-4 guessing entity "Yoga Mat". Following the question in red, all top guesses focus on plastic.