# OpenReview forum: "The Entity-Deduction Arena: A playground for probing the conversational reasoning and planning capabilities of LLMs"
_ICLR.cc/2024/Conference — Submitted to ICLR 2024_

### Official Review · Reviewer_TYnB · 2023-10-29

**Soundness:** 3 good
**Presentation:** 3 good
**Contribution:** 2 fair
**Rating:** 6
**Confidence:** 3

**Summary:**

The paper addresses the unreliability of Large Language Models (LLMs) when responding to ambiguous queries and emphasizes the need for intelligent agents that can ask clarifying questions. Using an entity-deducing game to evaluate LLMs' conversational skills, the authors find significant performance variations, with GPT-4 outperforming humans. The study also explores Behavior Cloning for model emulation and uses Reinforcement Learning to improve Vicuna models, aiming to better equip autonomous agents for ambiguous situations.

**Strengths:**

1. The paper underscores the need for intelligent conversational agents to proactively ask questions in uncertain situations, aiding effective problem-solving across various applications.
2. It highlights the shift from modular task management to employing Large Language Models (LLMs) for end-to-end autonomous agent development, enhancing complex task completion.
3. The paper tackles the challenge of accurately capturing nuanced user intents, suggesting the use of entity-deducing games to evaluate and improve LLMs' conversational skills.
4. A systematic evaluation of different LLMs is conducted, exploring enhancements using high-performing models and introducing methods to improve LLMs using game-based learning.

**Weaknesses:**

1. The major concern is that entity-deducing game is one small problem in terms of conversational reasoning and planning capabilities of language models. I am not convinced that a single game is a full reflection of the LLM reasoning ability
2. The experimental scale in the paper is limited. The conclusion in the paper should be drawn from at least thousands of samples.

**Questions:**

N/A

---

> ### Author Response · Authors · 2023-11-17
> **Author response**
>
> We are grateful to the reviewer for their valuable feedback and the insights provided. Below, we have addressed the questions and concerns raised:
>
> **Entity-deducing game is one small problem**: We completely agree that the benchmark we used does not fully reflect the reasoning ability of the LLM. Nevertheless we thinkg this problems adds to the well needed suite of test cases for measuring these characteristics of LLMs. We will revise the paper to make this point clearer. Please refer to our general response.
>
> **Experimental Scale**: We thank the reviewer for the suggestion. Please refer to the general questions for more discussion.

---

> > ### Comment · Reviewer_TYnB · 2023-11-21
> > **Raise the score to boardline accept**
> >
> > After reading the author's response, I now raise score from 5 to 6

---

### Official Review · Reviewer_HCV6 · 2023-11-01

**Soundness:** 2 fair
**Presentation:** 3 good
**Contribution:** 3 good
**Rating:** 5
**Confidence:** 4

**Summary:**

This paper proposes to evaluate the ability for LLMs to generate useful clarification questions in dialogue via an entity deduction game (20 Questions). The paper evaluates several popular LLMs as well as human performance, performs some analysis and experiments (e.g., on the game strategies, performance in planning and reasoning), and also experiments with distilling policies from strong LLMs to smaller open-source models.

**Strengths:**

The motivation is compelling and the task is clear. 20 Questions has been studied in prior work in the NLP community, but this seems to be the first large-scale study evaluating LLMs on the task. The experiments are also relatively clearly defined. I found research questions well-motivated and generally set up well, and they brought a lot of interesting experiments to mind that would be cool to explore more in more detail: e.g., could you somehow probe the top-N guesses directly from the model activations by training some additional probe on them? how are improvements on open-source models attributed to reasoning vs. planning (e.g. comparing the fine-tuned open source models to GPT-4 like the experiment in RQ2)?

**Weaknesses:**

* Some of the discussion is over-anthropomorphizing models. For example, in the introduction "intelligent behavior has been achieved" through what are common components of an enterprise dialogue system... it would be more precise and accurate to say that this is just traditionally how the problem of building dialogue systems, including ones that ask clarification questions, has been broken down. Some of the conclusions on why performance is strong is unsubstantiated, e.g., that GPT-4's performance is due to its zero-shot generalization ability. Wording like "realize" (in RQ1) also is attributing cognition to these models that I personally don't think is warranted.
* There needs to be precise, quantifiable evaluation of the judge's performance. Without this evaluation it's impossible for a reader to understand how much of an influence its errors has on the game.
* The evaluation set is really small! It's hard to conclude general performance on asking clarification questions when the set of target entities is so small. Ideally, we would be able to analyze performance across a number of entity features: how rare it is, how prototypical it is, how precise of an entity it is (e.g., specific breed of dog vs. dog in general), whether it's abstract or concrete, etc. With just 30 entities, it's hard to make any conclusions along these axes of difficulty. Figure 2 starts to get at this, and I think there are hints at interesting findings here (even just looking at human performance, to be honest) -- for example, umbrellas and cookie cutters are non-prototypical instances of their hypernyms in wordnet (respectively "canopy" and "kitchen utensil"). But with only 30 evaluation examples I don't think any strong conclusions could be made.
* Details on recruiting and managing crowdsourcing for the human baseline needs to be included in the main paper. There are also no details I could find in the appendix, e.g., on pay or the crowdsourcing platform used. There appears to be a lot of noise in the human dialogues (e.g., questions in the bottom game of table 6 appear to be obviously non-optimal) and I am wondering how workers were incentivized to actually try at the game. I also think analysis comparing the human strategies and LLM strategies should be in the main paper.
* I would have liked to see evaluation of human players on the reasoning task given a dialogue history generated by a model (and vice versa); essentially the experiment in Table 4 but replacing Vicuna 7B with a human. Since humans appear to perform worse, is this because they are worse at planning or reasoning (as defined in this paper)?
* I think the finding that these models have consistent strategies in question asking is cool, but it's not very surprising, because these are (mostly) deterministic models after all.

**Questions:**

* How were the entities chosen? The paper mentions they are manually curated, but from where?
* Why are the responses different for the celebrity dataset? ("dunno" vs. "maybe")
* What does "retrospective perspective" mean in Table 3 caption?
* How do you measure uncertainty in the model's top-N predictions as mentioned in the discussion of RQ1?

**Details Of Ethics Concerns:**

I could not find any details about crowdsourcing management, e.g. pay / incentives, recruiting, etc.

---

> ### Author Response · Authors · 2023-11-18
> **Author response**
>
> We thank the reviewer for the detailed and insightful feedback. In response, we have addressed the comments as follows:
>
> **Over-anthropomorphizing**: We appreciate and agree with the reviewer's viewpoint and will tweak the language carefully to avoid the accidental over-anthropomorphization in our writing style. Note that we were not suggesting that "intelligent behavior" is human-like or human-level but we can see in retrospect how our presentation might be construed that way -- a careful review and rephrasing of parts of the paper should alleviate this.
>
> **Evaluating the Judge**: We acknowledge the need for better quantification of the errors made by Judge. Since obtaining the ground truth is challenging, we decided to conduct a human evaluation of Judge (GPT3.5-turbo-0613) responses. This evaluation involved non-paid volunteers from our institute who assessed the correctness of Judge's responses to questions asked given the oracle entity or celebrity name. Due to time constraints during the rebuttal, based on our current annotation progress, we were only able to evaluate 223 Judge responses on Things and 237 Judge responses on Celebrities. The human evaluators identified 7 out of 223 (3.13%) and 7 out of 237 (2.95%) responses as problematic or incorrect. A more comprehensive summary of these results will be included in our revision.
>
> **The evaluation set is small**: We agree that having a larger evaluation dataset would enable more comprehensive and detailed analysis at a finer granularity. To this end, we are currently working on expanding the evaluation by incorporating 300 examples for each dataset, across all the models being compared. We will incorporate the results of this evaluation in our general response.
>
> **Human evaluation details**: We have discussed some details of creating the human baseline in the Appendix D. We agreed that some contents needed to be moved to the main text, while some information needed to be supplemented. In this study, we did not hire any paid crowd-source workers. As discussed in Section 3, because the complexity of the multi-turn game setup, we developed an internal demo server (shown in figure 4) and relied solely on enthusiastic internal volunteers who were interested in our study. Following a 30-day trial period, 108 volunteers contributed to this study, and we successfully collected a total of 216 human game play sessions for Things and Celebrities.
>
> To ensure the quality of the data, we manually inspected human game plays to filter out low quality data. The authors reviewed all human game play sessions, and removed sessions that do not complete the entity deduction task, contain irrelavant chit-chat, or contain nonsensical repeated questions, leaving only sensible game plays. In the end 201 sessions were selected as valid game play.
>
> We appreciate the suggestion and will incorporate a thorough discussion comparing the strategies employed by humans and LLMs into the main text.
>
> **Evaluation of Human players on reasoning tasks**: We are currently conducting some additional experiments and will provide supplementary results shortly.
>
> **Deterministic models**: In our study, we compute statistics for each game by performing 5 repetitions for each game. We apply a sampling approach and use a temperature of 0.8 for the guesser models throughout our study. We observed that the gameplay can vary significantly, resulting in completely different outcomes. For example, the failure results presented in Table 2, 8, 12 are derived from the same model, entity, and settings.

---

> > ### Author Response · Authors · 2023-11-18
> > **Author response to the questions**
> >
> > **How were the entities chosen**: We manually created several categories and then asked GPT-3.5 to generate additional categories. Next, we asked it to generate unique entities for these categories without repetition. We will provide the prompts for creating these entities and celebrities' names in our revision.
> >
> > **Why the Celebrities dataset use "Dunno"**: The Things dataset contains entities that has many different forms, such as if asking "is this made by metal?" for the entity "Chopstick". The answer to this question could be "Maybe" because sometimes they are made by metal and sometimes are not. On the other hand, in the Celebrities dataset, most questions can typically be clearly answered with a yes or no. These questions often refer to gender, birth year, or other specific information about a person. However, there is a possibility that the model might not know the answer to certain questions, such as "Does this celebrity smoke?". In these cases, the answer could be "Dunno" as it is considered very personal information. As a general observation, we have found that less than 1% of the responses in the Celebrities dataset are "Dunno", while around 20% of the responses in the Things dataset are "Maybe". Note that the percentage of "Dunno" or "Maybe" responses can vary depending on the specific Guesser model being used.
> >
> > **Meaning of Retrospective perspective**: We used strikethrough lines to indicate the guesses that were inconsistent with the Judge's answer, based on our knowledge and evaluation. This visual representation illustrates the authors' thought process to identify the irrelevant guesses after obtaining the Judge's response. Note that we observed a strong correspondence between our assessment of irrelevant guesses and the model's top predictions. This suggests that the model may be capable of similar reasoning to eliminate certain guesses. However, this observation has not been quantitatively verified yet. We will clarify these points in our paper.
> >
> >
> > **Measurement of uncertainty**: We asked the Guesser model, "On a scale of 1-5, how confident are you in your top guesses based on the current conversation history?" We observed some qualitative correlation between the model's predicted uncertainty and significant shifts in its top guesses. However, this pattern was mainly observed in more powerful models such as GPT-4. In the case of weaker models like Vicuna, the top guesses or confidence scores generated by the model were not very sensible. We will include these specific details in the paper to enhance its clarity.

---

> > > ### Comment · Reviewer_HCV6 · 2023-11-22
> > >
> > > > We manually created several categories and then asked GPT-3.5 to generate additional categories. Next, we asked it to generate unique entities for these categories without repetition.
> > >
> > > Hm, I worry about this decision -- in particular because if GPT-3.5 is already giving high prior probability to the items it's guessing, it would probably be better at guessing them than other plausible items within the categories that it may be less familiar with. Did you experiment with evaluating on items from these categories that are more adversarially chosen?

---

> > > > ### Author Response · Authors · 2023-11-23
> > > >
> > > > > Hm, I worry about this decision -- in particular because if GPT-3.5 is already giving high prior probability to the items it's guessing, it would probably be better at guessing them than other plausible items within the categories that it may be less familiar with. Did you experiment with evaluating on items from these categories that are more adversarially chosen?
> > > >
> > > > We believe this point is valid and the items generated by the models may be easier for them to guess. We indeed attempted to manually come up with 10 items that were not included in the datasets, and it's true that these items performed worse compared to the entities generated by the models. This could be due to the reason mentioned by the reviewer, or it could simply be because the items we came up with were less commonly known. Since we actually used GPT-3.5 to generate thousands of unique items, it becomes quite challenging to think of something that has not been covered in the generated list. We will further investigate the issue and conduct additional analysis in our revision to compare the impact of using human-selected or web-crawled items versus model-generated ones.

---

> ### Author Response · Authors · 2023-11-23
> **Author response (Cont'd)**
>
> **Evaluation of Human players on reasoning tasks**:
>
> We thank the reviewer for the valuable suggestion. While we are still awaiting the full results, we would like to share our current findings below:
>
>
> | Approach | Things | Celeb |
> |--------------------------|---------------|------------|
> | GPT-4 | 0.40±0.05 | 0.48±0.03 |
> | Vicuna 7B | 0.10±0.04 | 0.04±0.03 |
> | Human | 0.20±0.04 | 0.25±0.03 |
> | Vicuna 7B → Human | 0.1 | |
> | GPT-4 → Human | 0.14 | |
> | Human → Vicuna 7B | 0.11±0.04 | 0.03±0.01 |
> | Human → GPT-4 | 0.15±0.01 | 0.18±0.06 |
>
> Note that these results are based on the evaluation set in the submission for consistency. In our next update, we plan to extend the study to include a new dataset with 300 items/names. Following the setup outlined in the paper, only the last turn for each dialog trajectory is replaced. We are currently relying on non-paid volunteers for the annotation.
>
> The findings indicate that when GPT-4 is employed for the last step reasoning in human trajectories, there is a decrease in performance. One possible explanation for this is that while GPT-4 may excel in comprehending model-generated trajectories, it may struggle to fully interpret the planning trajectory of humans based on the observed turns. This suggests that models may be more compatible with model-generated trajectories rather than human trajectories. For instance:
>
> | Turn | Human Question            | Judge Answer | GPT-4 Question   |
> |------|---------------------------|--------------|------------------|
> | 1    | Is it a VR Headset        | No.          |                  |
> | 2    | Is it made of metal       | Yes.         |                  |
> | 3    | Is it used at home?       | Yes.         |                  |
> | 4    | Is it a chair?            | No.          |                  |
> | 5    | Is it a table?            | No.          |                  |
> | 6    | Is it a faucet            | No.          |                  |
> | 7    | Is it a fridge?           | No.          |                  |
> | 8    | Is it a bed frame?        | No.          |                  |
> | 9    | Is it a window frame?     | No.          |                  |
> | 10   | Is it a stationary        | Yes.         |                  |
> | 11   | Is it a pen               | No.          |                  |
> | 12   | Is it a pencil box        | No.          |                  |
> | 13   | Is it a scisor            | No.          |                  |
> | 14   | Is it a letter opener?    | No.          |                  |
> | 15   | Does it write?            | Yes.         |                  |
> | 16   | Is it a metal pen         | No.          |                  |
> | 17   | Is it a mechanical pencil | No.          |                  |
> | 18   | Is it a printer           | Bingo!       | Is it a Stapler? |
>
> Even though it was previously established that the entity can write, GPT-4 still asked if the entity is a Stapler or not.

---

### Official Review · Reviewer_DwoQ · 2023-11-01

**Soundness:** 2 fair
**Presentation:** 3 good
**Contribution:** 2 fair
**Rating:** 3
**Confidence:** 5

**Summary:**

This paper, based on the prototype of the entity deduction game, carries out a series of experimental designs. The author presents the purpose of the experiment, the process, defines the criteria for success or failure, and introduces a scoring formula for evaluation. In datasets related to entities and things, a comparison experiment of the entity deduction game was conducted on several LLMs. The results showed that GPT-4 possesses a superior ability to narrow down compared to other LLMs. Furthermore, the paper explores the possibility of enabling other weaker models to acquire this capability through reinforcement learning.

**Strengths:**

The perspective of the paper is quite interesting. It identifies specific characteristics of GPT-4 in the entity deduction game task, such as superior LLM knows about backtracking or employing a good strategy to divide the potential solution space instead of simply performing enumeration.

**Weaknesses:**

In the presentation, I believe the author should use a paragraph to summarize the unique contributions of the paper.
1. The paper lacks novelty and robustness to pass the bar of ICLR. The experiments designed don't show much improvement compared with referenced prototype. The datasets leveraged in the experiments, both in terms of range (two categories entities and things) and quantity, have certain limitations. However, the paper lacks discussion on this aspect. Moreover, given the inherent randomness in GPT's responses, it raises the question of whether multiple repeated experiments were conducted to ensure the stability of the results.
2. The formula introduced in this paper lacks a thorough explanation and rigor. Why pick linearity? Any insight to pick 0.02 as coefficient?
3. The observation and conclusion from experiments seem rather preliminary. The "entity deduction game" and the so-called "path planning" requires a more strict formulation for deeper analysis. This is especially true for the criterion of what can be a good path.
4. The ability of path planning can easily be influenced by domain knowledge. I suggest that if we want to rigorously examine planning capability, perhaps we could consider a much narrower and common sense domain, for example, having GPT guess numbers within a certain range.
PS: I  did try this number-guessing experiment on GPT-4. GPT-4 doesn't necessarily choose the binary search as the optimal path; it often opts for a narrower range enumeration.

**Questions:**

As stated above

---

> ### Author Response · Authors · 2023-11-17
> **Author response**
>
> We thank the reviewer's comments and feedback on our work. Next we address the questions and concerns:
>
> **Not enough improvement over referenced LLM**: Perhaps we misunderstood the reviewer, but we feel that our methods show strong improvement over our baselines in Table 1 -- both BC trained models (V-FT* models in Table 1 -- e.g. V-FT 7B) and RL trained models (V-RLGP* models in Table 1 -- e.g. V-RLGP 7B) show more than 100% improvement over the base model (Vicuna 7B). Perhaps the reviewer is suggesting that we compare our trained models against GPT-4 ? In that case it would be unfair since the GPT-4 model is much larger than the models we trained.
>
> **Not enough diversity in the datasets and no discussion**: In section 3.1 and Appendix A, we discussed that we aim to create rich and diverse datasets. In fact, our datasets are diverse. The **Things** dataset encompasses a wide range of categories, such as objects, animals, foods, plants, vehicles, clothing, professions, materials, instruments, places, birds, sports, buildings, furniture, celestial bodies, mythical creatures, games, body parts, beverages, weather phenomena, groups, gemstones, toys, tools, patterns, appliances, and microorganisms, among others.
>
> The **celebrities** dataset meanwhile includes names from different nationalities, eras of life, and various occupations, including entertainment, sports, politics, entrepreneurship, science, philanthropy, writing, music, military, engineering, and more. We used 980 entities for the Things dataset and 133 entities for celebrities. Additionally, we are in the process of collecting and evaluating 300 additional examples for each dataset to further expand its size. The results of this effort will be shared in our general response.
>
> **No repeated experiment**: We apologize if we didn't emphasize this enough in the paper, but we did in fact repeat our experiments. In Table 1, our main results were obtained by conducting 5 repetitions for each gameplay, which provided the standard deviation as shown in the paper. This was further discussed in Section 4.
>
> **Why linearity in reward design**: It is commonplace in the RL community to use rewards based on heuristics that assign numeric values commensurate with desriable properties, and then combine these scores linearly. For instance, similar heuristics have been used in controlling the agent's movements [4] and in task-oriented chatbots [5]. In this paper, the score we proposed combines a reward for a good game outcome with a penalty for taking longer to achieve the right answer and 0.02 is just a hyperparameter that balances these two properties. With a value of 0.02 the model is still rewarded 0.7 for answering a question correctly after the full 20 steps. Values greater than 0.02 would reduce the rewards for games that take longer, while smaller values would penalize the model less for taking a longer time. In our early exploratory experiments we found 0.02 to lead to models with higher final success rate, compared to other values. Admittedly, reward selection is an art in RL, rather than a well understood science and it is possible that there are better choices out there, both in the heuristics chosen, and how their values are computed and combined -- we would be happy to add such qualifications to the paper.
>
> **Lack path planning analysis**: In fact, we provide a comprehensive qualitative analysis of what constitutes a good or bad path for the game. In Table 2 and Tables 8, 9, and 12, we have highlighted how the models are different in their planning ability and illustrated and explained the characteristics of good "path planning" and identify the common failure modes that lead to inadequate planning. The criteria for a good path indeed depends on the domain and may be hard to generalize at this stage of our understanding of LLMs. However, we discussed some general observations. In Table 3 and 13, we present the criteria for successful planning, which include 1) prioritizing high-level questions before addressing specific details and enumerations, 2)being aware of the current state and asking questions to effectively bi-partition the search space, and 3) being able to recognize when the current path is incorrect and occasionally backtracking to consider previously overlooked options. All of these aspects are discussed in Section 4 of our paper, and we will make them clearer.
>
> [4] Jens, et al. *"Reinforcement learning in robotics: A survey."* The International Journal of Robotics Research 32.11 (2013)
>
> [5] Bhuwan, et al. *"Towards end-to-end reinforcement learning of dialogue agents for information access."* ACL (2016).

---

> > ### Author Response · Authors · 2023-11-17
> > **Author response (Cont'd)**
> >
> > **Comments regarding domain knowledge**: We commend the reviewer's diligence in exploring the range guessing game. We think that it is a much simpler setup which may not adequately address some of the questions we are raising about the state of the art LLMs. As far as domain knowledge goes, even in the range guessing game each model may have a varying level of understanding of arithmetic, such as determining which number is larger or smaller and thus disentangling knowledge from planning would be a problem even in that set up. Domain knowledge is indeed a complicating factor but we believe that it would be difficult to entirely disentangle the knowledge and planning and remove the knowledge factor in any set up. In fact, the goal of our set up is to test a model's ability to **handle** the ambiguity in its domain-knowledge with better planning. We chose a real world game on common items because most humans are able to handle this problem intuitively without any specialized training in a domain like mathematics and wanted to explore how LLMs perform comparatively on this.
> >
> > At the same time, we did indeed make an attempt to explore how domain knowledge plays a role in this set up. In section 4 and Appendix E, we discuss how stronger models possess a higher level of domain knowledge compared to humans. It is important to note that our focus is on the Guesser role, which aim to guess rather than provide accurate answers. However, it is undeniable that a stronger model will have a more comprehensive taxonomy representation of entities in its intrinsic knowledge store. We considered this an essential aspect of the planning ability, as we believe that the ability to plan is closely linked to a deep comprehension, digestion and representation of knowledge.

---

### Official Review · Reviewer_oYKG · 2023-11-01

**Soundness:** 3 good
**Presentation:** 3 good
**Contribution:** 3 good
**Rating:** 8
**Confidence:** 3

**Summary:**

This research assesses the reasoning performance of Large Language Models (LLMs) using the Entity-Deduction Arena (EDA) task, in which LLMs ask a judge questions to infer an entity. According to the study, GPT-4 performs better than humans when it comes to strategic questioning. It emphasizes how well Reinforcement Learning (RL) and Behavior Cloning (BC) work to improve reasoning abilities, especially in larger models, and how well BC transfers skills from more complex models to simpler ones. The study enhances LLMs' capacity to handle unclear queries by indicating that LLMs have an underlying structure of knowledge that may be improved with training.

**Strengths:**

1. Innovative Testbed: The research presents the EDA as a novel testbed for assessing LLMs' strategic planning and deductive reasoning skills in handling unclear user intents through questioning to deduce entities.

2. Performance Analysis: Systematically assessing different LLMs, the study finds notable variations in their performance, with more powerful models such as GPT-4 surpassing human players, highlighting the sophisticated capabilities of existing LLMs.

3. Model Size Insights: Results imply that performance increase may not be primarily determined by the model's size. This disproves the notion that larger models are necessarily more effective and shows that smaller models can also gain a great deal from fine-tuning.

**Weaknesses:**

The paper could discuss more extensively the autonomous learning capabilities of LLMs. Prior research suggests LLMs like GPT-4 may have limited autonomous planning capacity and rely heavily on well-designed heuristics. The paper's task relies solely on textual goals, which may not fully capture the challenges LLMs face with numerical or spatial reasoning, and how they adapt to diverse prompt requirements

**Questions:**

The paper observes that LLMs tend to fall into repetitive patterns and accumulate errors, particularly in weaker models. Can the authors elaborate on potential approaches to mitigate these undesirable behaviors? Additionally, how might these patterns impact the long-term learning and adaptability of LLMs in more complex or dynamic environments?

---

> ### Author Response · Authors · 2023-11-17
> **Author response**
>
> We thank the reviewer for providing us with constructive feedback and supportive comments! We address the questions in below:
>
> **Numerical or spatial reasoning**: Please also refer to our general response. We acknowledge that this paper does not address numerical or spatial reasoning, and that was intentional. We will make it clearer that we are only evaluating specific capacities of LLMs involving deductive reasoning, state tracking, fact recalling and strategical planning, under multi-turn information querying scenarios. We really like the suggestion to expand our evaluation to include more realistic tasks (e.g., spatial planning, numerical reasoning, tool-use,  and task-completion chatbot) beyond text-only objectives. To achieve that, we believe that a controlled testing environment with clear metrics and evaluation would be essential as a first step towards generalizing to diverse scenarios and prompt requirements.
>
> **Mitigation of undesirable behavior**: In our experiments, we have observed that behavior cloning (BC) training can effectively mitigate undesirable behavior and generation artifact we showed in Table 9, such as repetition. We think that the model's confidence level might play a crucial role in this mitigation. When the model is uncertain about which action to take (we can probe the model by asking its current prediction and confidence level), we have noticed a tendency for it to exhibit degeneration. By incorporating reasoning and planning capabilities learned from stronger models into weaker models through BC, we can demonstrate good trajectory and enable the model to follow a more effective strategy in generating questions. Generally, we consider these issues to be fundamental to autoregressive training via teacher forcing and believe that addressing them is essential for the effective learning and adaptability of LLMs. As far as we know, mitigating these issues for LM remains an open question for further exploration [2, 3].
>
> [2] Ari Holtzman, Jan Buys, Li Du, Maxwell Forbes, and Yejin Choi. *The curious case of neural text degeneration*. In ICLR, 2019.
>
> [3] Jin Xu, Xiaojiang Liu, Jianhao Yan, Deng Cai, Huayang Li, and Jian Li. *Learning to break the loop: Analyzing
> and mitigating repetitions for neural text generation*. In NeurIPS, 2022.

---

### Author Response · Authors · 2023-11-17
**General response**

We thank all reviewers for their valuable feedback and insightful comments. In the following, we will address some of the common concerns and questions:

**EDA does not fully reflect the LLM capability**: Various benchmarks cover general reasoning (e.g., HELM) and more specific math (e.g., GSM8K), spatial (e.g., assessment suite[1]) and commonsense reasoning (e.g., ARC) aspects for LLM evaluation. As mentioned in the related work section, our goal is to introduce a benchmark that brings a unique perspective to **complement** existing benchmarks, rather than replacing any of them. The main focus of this paper is to take the initial step in evaluating the model's capacity involving deductive reasoning, state tracking, fact-recalling and strategical planning, under **multi-turn information querying scenarios**. We think setting up benchmark for assessing (and further improving) these aspects are novel and critical towards building autonomous agents that proactively lead the conversation. We will revise our paper and modify the language to make this clearer.


**Dataset size is not sufficient**: We are working on the additional experiments. Will update this shortly.

[1] Karthik Valmeekam, Alberto Olmo, Sarath Sreedharan, and Subbarao Kambhampati. *Large language models still can’t plan (a benchmark for llms on planning and reasoning about change).* arXiv preprint arXiv:2206.10498, 2022.

---

> ### Author Response · Authors · 2023-11-23
> **General response (Cont'd)**
>
> **Dataset size is not sufficient**: We acknowledge this limitation, and have since expanded our dataset size from 30 things and 30 celebrities to 300 each, containing 600 entities in total that are not overlapping with the training datasets. Specifically, while expanding things dataset, entities were chosen from the various categories including objects, animals, foods, plants, vehicles, clothing, professions, materials, instruments, places, birds, sports, buildings, furniture, celestial bodies, mythical creatures, games, body parts, beverages, weather phenomena, groups, gemstones, toys, tools, patterns, appliances, and microorganisms. For celebrities datasets, the additional datasets cover names from different nationalities, eras of life, and various occupations. The updated experiment results can be found below:
>
> |                  | Things |         |       |       | Celebrity |         |       |       |
> |------------------|--------|---------|-------|-------|-----------|---------|-------|-------|
> |                  | #Turns | Success | #Yes  | Score | #Turns    | Success | #Yes  | Score |
> |  GPT-4           | 18.548 | 0.227   | 5.91  | 0.188 | 17.16     | 0.56    | 6.49  | 0.449 |
> | GPT-3.5          | 19.104 | 0.157   | 6.268 | 0.128 | 19.137    | 0.21    | 6.59  | 0.164 |
> | Claude-2         | 19.247 | 0.12    | 4.809 | 0.099 | 17.723    | 0.35    | 5.057 | 0.291 |
> | Claude-1         | 19.365 | 0.107   | 4.425 | 0.088 | 17.823    | 0.367   | 5.007 | 0.3   |
> | Vicuna 13B       | 19.263 | 0.12    | 5.083 | 0.099 | 19.217    | 0.173   | 5.893 | 0.137 |
> | Vicuna 7B        | 19.743 | 0.043   | 5.807 | 0.035 | 19.83     | 0.03    | 5.29  | 0.024 |
> | V-FT 7b (All)    | 19.408 | 0.1     | 6.401 | 0.082 | 19.717    | 0.083   | 7.433 | 0.064 |
> | V-FT 7b (Things) | 19.589 | 0.074   | 6.124 | 0.06  | 19.75     | 0.073   | 6.713 | 0.056 |
> | V-FT 7b (Celebs) | 19.916 | 0.013   | 1.672 | 0.011 | 19.767    | 0.07    | 6.277 | 0.054 |
>
> Overall, it is observed that the performance on the new evaluation datasets decreased for all methods, which is likely attributed to the increased difficulty of the evaluation set. The relative comparison between different methods largely remains unchanged, except for Claude-1 which slightly outperforms Claude-2. Owing to time limitations and the larger number of evaluation items, we were only able to conduct one repetition for each entity or name. We are currently in the process of collecting complete results with five repetitions and other models in the paper. Once they are available, we will revise the paper accordingly.

---

### Meta-Review · Area_Chair_Qn2W · 2023-12-06

**Metareview:**

This paper presents an "entity deduction" task, where a model has to ask questions to determine the identity of an entity that it does not know.  This is framed as a test for conversational reasoning and planning capabilities. The paper presents two datasets, Things and Celebrities, with evaluation sets of 300 examples (post-rebuttal) and larger training sets. GPT-3.5 turbo is used as the "judge" to answer questions for the "guesser" LLM.  The paper experiments with closed-source models as well as distilling those models into open-source models.

The reviewers found this to be an interesting task with clear motivation. It spawns a number of interesting research questions for future work (HCV6).  The evaluation of different LLMs on the task was interesting.

The critiques by the reviewers generally reflected that this paper's benchmark is a bit narrow.  The reviewers perceived a lack of generality in the task setting (HCV6). The authors pointed to section 3.1 and Appendix A to support this, but while there's diversity in terms of entities, the dataset is nevertheless focused on fairly specific object categories (e.g., half the celebrities are actors/actresses or singers). The larger size of the evaluation sets may expand this a bit, but it does not change the overall domains of reasoning here. As HCV6 says, this actually has quite a major impact on the conclusions drawn from the benchmark, as the experiments reported in the respo
nse period show.

Furthermore, HCV6 points out past work on systems to play 20 questions, although this paper is the first to address it from the LLM perspective.

The decision for this paper was calibrated against a number of other borderline papers taking reviewer comments into account, and was not heavily influenced by the low-scoring review here.

**Justification For Why Not Higher Score:**

Both the 8 and the 3 are weak reviewers. I am placing the most weight on HCV6 and I agree with those critiques; see my meta-review. The paper is too narrow to form a strong contribution, although the ideas here are quite valuable.

**Justification For Why Not Lower Score:**

N/A

---

### Decision · Program_Chairs · 2024-01-16

Reject